# SPIDER: Near-Optimal Non-Convex Optimization via Stochastic Path Integrated Differential Estimator

**Cong Fang**[1] *  **Chris Junchi Li**[2]  **Zhouchen Lin**[1][†]  **Tong Zhang**[2]

[1]Key Lab. of Machine Intelligence (MoE), School of EECS, Peking University

[2]Tencent AI Lab

{fangcong, zlin}@pku.edu.cn    junchi.li.duke@gmail.com    tongzhang@tongzhang-ml.org

## Abstract

In this paper, we propose a new technique named *Stochastic Path-Integrated Differential EstimatoR* (SPIDER), which can be used to track many deterministic quantities of interests with significantly reduced computational cost. Combining SPIDER with the method of normalized gradient descent, we propose SPIDER-SFO that solve non-convex stochastic optimization problems using stochastic gradients only. We provide a few error-bound results on its convergence rates. Specially, we prove that the SPIDER-SFO algorithm achieves a gradient computation cost of $\mathcal{O}\left(\min(n^{1/2}\epsilon^{-2}, \epsilon^{-3})\right)$ to find an $\epsilon$-approximate first-order stationary point. In addition, we prove that SPIDER-SFO nearly matches the algorithmic lower bound for finding stationary point under the gradient Lipschitz assumption in the finite-sum setting. Our SPIDER technique can be further applied to find an $(\epsilon, \mathcal{O}(\epsilon^{0.5}))$-approximate second-order stationary point at a gradient computation cost of $\tilde{\mathcal{O}}\left(\min(n^{1/2}\epsilon^{-2} + \epsilon^{-2.5}, \epsilon^{-3})\right)$.

## 1 Introduction

In this paper, we study the optimization problem

$$\underset{\mathbf{x}\in\mathbb{R}^d}{\text{minimize}} \quad f(\mathbf{x}) \equiv \mathbb{E}\left[F(\mathbf{x}; \boldsymbol{\zeta})\right] \tag{1.1}$$

where the stochastic component $F(\mathbf{x}; \boldsymbol{\zeta})$, indexed by some random vector $\boldsymbol{\zeta}$, is smooth and possibly *non-convex*. Non-convex optimization problem of form (1.1) contains many large-scale statistical learning tasks and is gaining tremendous popularity due to its favorable computational and statistical efficiency [5–7]. Typical examples of form (1.1) include principal component analysis, estimation of graphical models, as well as training deep neural networks [17]. The expectation-minimization structure of stochastic optimization problem (1.1) allows us to perform iterative updates and minimize the objective using its stochastic gradient $\nabla F(\mathbf{x}; \boldsymbol{\zeta})$ as an estimator of its deterministic counterpart.

A special case of central interest is when the stochastic vector $\boldsymbol{\zeta}$ is finitely sampled. In such *finite-sum* (or *offline*) case, we denote each component function as $f_i(x)$ and (1.1) can be restated as

$$\underset{\mathbf{x}\in\mathbb{R}^d}{\text{minimize}} \quad f(\mathbf{x}) = \frac{1}{n}\sum_{i=1}^{n} f_i(\mathbf{x}) \tag{1.2}$$

[†]Corresponding author.

where $n$ is the number of individual functions. Another case is when $n$ is reasonably large or even infinite, running across of the whole dataset is exhaustive or impossible. We refer it as the *online* (or *streaming*) case. For simplicity of notations we will study the optimization problem of form (1.2) in both finite-sum and online cases till the rest of this paper.

One important task for non-convex optimization is to search for, given the precision accuracy $\epsilon > 0$, an *$\epsilon$-approximate first-order stationary point* $\mathbf{x} \in \mathbb{R}^d$ or $\|\nabla f(\mathbf{x})\| \leq \epsilon$. In this paper, we aim to propose a new technique, called the *Stochastic Path-Integrated Differential EstimatoR* (SPIDER), which enables us to construct an estimator that tracks a deterministic quantity with significantly lower sampling costs. As the readers will see, the SPIDER technique further allows us to design an algorithm with a faster rate of convergence for non-convex problem (1.2), in which we utilize the idea of *Normalized Gradient Descent* (NGD) [18, 26]. NGD is a variant of Gradient Descent (GD) where the stepsize is picked to be inverse-proportional to the norm of the full gradient. Compared to GD, NGD exemplifies faster convergence, especially in the neighborhood of stationary points [25]. However, NGD has been less popular due to its requirement of accessing the full gradient and its norm at each update. In this paper, we estimate and track the gradient and its norm via the SPIDER technique and then hybrid it with NGD. Measured by *gradient cost* which is the total number of computation of stochastic gradients, our proposed SPIDER-SFO algorithm achieves a faster rate of convergence in $\mathcal{O}(\min(n^{1/2}\epsilon^{-2}, \epsilon^{-3}))$ which outperforms the previous best-known results in both finite-sum [3][32] and online cases [24] by a factor of $\mathcal{O}(\min(n^{1/6}, \epsilon^{-0.333}))$.

For the task of finding stationary points for which we already achieved a faster convergence rate via our proposed SPIDER-SFO algorithm, a follow-up question to ask is: *is our proposed SPIDER-SFO algorithm optimal for an appropriate class of smooth functions?* In this paper, we provide an *affirmative* answer to this question in the finite-sum case. To be specific, inspired by a counterexample proposed by Carmon et al. [10] we are able to prove that the gradient cost upper bound of SPIDER-SFO algorithm matches the *algorithmic lower bound*. To put it differently, the gradient cost of SPIDER-SFO *cannot* be further improved for finding stationary points for some particular non-convex functions.

## 1.1 Related Works

In the recent years, there has been a surge of literatures in machine learning community that analyze the convergence property of non-convex optimization algorithms. Limited by space and our knowledge, we have listed all literatures that we believe are mostly related to this work. We refer the readers to the monograph by Jain et al. [19] and the references therein on recent general and model-specific convergence rate results on non-convex optimization.

**SGD and Variance Reduction** For the general problem of finding approximate stationary points, under the smoothness condition of $f(\mathbf{x})$, it is known that vanilla Gradient Descent (GD) and Stochastic Gradient Descent (SGD), which can be traced back to Cauchy [11] and Robbins & Monro [33] and achieve an $\epsilon$-approximate stationary point with a gradient cost of $\mathcal{O}(\min(n\epsilon^{-2}, \epsilon^{-4}))$ [16, 26].

Recently, the convergence rate of GD and SGD have been improved by the variance-reduction type of algorithms [22, 34]. In special, the finite-sum Stochastic Variance-Reduced Gradient (SVRG) and online Stochastically Controlled Stochastic Gradient (SCSG), to the gradient cost of $\tilde{\mathcal{O}}(\min(n^{2/3}\epsilon^{-2}, \epsilon^{-3.333}))$ [3, 24, 32].

**First-order method for finding approximate second-order stationary points** It has been shown that for machine learning methods such as deep learning, approximate stationary points that have at least one negative Hessian direction, including saddle points and local maximizers, are often *not* sufficient and need to be avoided or escaped from [12, 15]. Recently, many literature study the problem of how to avoid or escape saddle points and achieve an $(\epsilon, \delta)$-approximate second-order stationary point $\mathbf{x}$ at a polynomial gradient cost, i.e. an $\mathbf{x} \in \mathbb{R}^d$ such that $\|\nabla f(\mathbf{x})\| \leq \epsilon, \lambda_{\min}(\nabla^2 f(\mathbf{x})) \geq -\delta$ [1, 2, 4, 8, 15, 18, 20, 21, 23, 25, 30, 31, 35, 38]. Among them, the group of authors Ge et al. [15], Jin et al. [20] proposed the noise-perturbed variants of Gradient Descent (PGD) and Stochastic

Gradient Descent (SGD) that escape from all saddle points and achieve an $\epsilon$-approximate second-order stationary point in gradient cost of $\tilde{\mathcal{O}}(\min(n\epsilon^{-2}, poly(d)\epsilon^{-4}))$ stochastic gradients. Levy [25] proposed the noise-perturbed variant of NGD which yields faster evasion of saddle points than GD.

The breakthrough of gradient cost for finding second-order stationary points were achieved in 2016/2017, when the two recent lines of literatures, namely FastCubic [1] and CDHS [8] as well as their stochastic versions [2, 35], achieve a gradient cost of $\tilde{\mathcal{O}}(\min(n\epsilon^{-1.5}+n^{3/4}\epsilon^{-1.75}, \epsilon^{-3.5}))$ which serves as the best-known gradient cost for finding an $(\epsilon, \mathcal{O}(\epsilon^{0.5}))$-approximate second-order stationary point before the initial submission of this paper.[3] [4] In particular, Agarwal et al. [1], Tripuraneni et al. [35] converted the cubic regularization method for finding second-order stationary points [27] to stochastic- gradient based and stochastic-Hessian-vector-product-based methods, and Allen-Zhu [2], Carmon et al. [8] used a Negative-Curvature Search method to avoid saddle points. See also recent works by Reddi et al. [31] for related saddle-point-escaping methods that achieve similar rates for finding an approximate second-order stationary point.

**Other concurrent works** As the current work is carried out in its final phase, the authors became aware that an idea of resemblance was earlier presented in an algorithm named the *StochAstic Recursive grAdient algoritHm* (SARAH) [28, 29]. Despite the fact that both our SPIDER-SFO and theirs adopt the recursive stochastic gradient update framework and our SPIDER-SFO can be viewed as a variant of SARAH with normalization, our work differ from their works in two aspects:

(i) Our analysis techniques are totally different from the version of SARAH proposed by Nguyen et al. [28, 29]. Their version can be seen as a variant of gradient descent, while ours hybrids the SPIDER technique with normalized gradient descent. Moreover, Nguyen et al. [28, 29] adopt a large stepsize setting (in fact their goal was to design a memory-saving variant of SAGA [13]), while our SPIDER-SFO algorithm adopt a small stepsize that is proportional to $\epsilon$. All these are essential elements of our superior achievements in convergence rates;

(ii) Our proposed SPIDER technique is a much more general variance-reduced estimation method for many quantities (*not* limited to gradients) and can be flexibly applied to numerous problems, e.g. stochastic zeroth-order method.

Soon after the initial submission to NIPS and arXiv release of this paper, we became aware that similar convergence rate results for stochastic first-order method were also achieved independently by the so-called SNVRG algorithm [39, 40]. SNVRG [40] obtains a gradient complexity of $\tilde{\mathcal{O}}(\min(n^{1/2}\epsilon^{-2}, \epsilon^{-3}))$ for finding an $\epsilon$-approximate first-order stationary point and achieves a $\tilde{\mathcal{O}}(\epsilon^{-3.5})$ gradient cost for finding an $(\epsilon, \mathcal{O}(\epsilon^{0.5}))$-approximate second-order stationary point [39]. By exploiting the third-order smoothness, an SNVRG variant can also achieve an $(\epsilon, \mathcal{O}(\epsilon^{0.5}))$-approximate second-order stationary point in $\tilde{\mathcal{O}}(\epsilon^{-3})$ stochastic gradient costs [39].

## 1.2 Our Contributions

In this work, we propose the Stochastic Path-Integrated Differential Estimator (SPIDER) technique, which significantly avoids excessive access of stochastic oracles and reduces the time complexity. Such technique can be potential applied in many stochastic estimation problems.

(i) We propose the SPIDER-SFO algorithm (Algorithm 1) for finding approximate first-order stationary points for non-convex stochastic optimization problem (1.2), and prove the optimality of such rate in at least one case. Inspired by recent works Carmon et al. [8, 10], Johnson & Zhang [22] and independent of Zhou et al. [39, 40], this is the *first* time that the gradient cost of $\mathcal{O}(\min(n^{1/2}\epsilon^{-2}, \epsilon^{-3}))$ in both upper and lower (finite-sum only) bound for finding first-order stationary points for problem (1.2) were obtained.

(ii) Following Allen-Zhu & Li [4], Carmon et al. [8], Xu et al. [38], we propose SPIDER-SFO$^+$ algorithm for finding an approximate second-order stationary point for non-convex stochastic optimization problem. To best of our knowledge, this is also the *first* time that the gradient cost of $\tilde{\mathcal{O}}(\min(n^{1/2}\epsilon^{-2} + \epsilon^{-2.5}, \epsilon^{-3}))$ achieved with standard assumptions. We leave the details of SFO in the long version of our paper: `https://arxiv.org/abs/1807.01695`

(iii) We propose a new and simpler analysis framework for proving convergence to approximate stationary points. One can flexibly apply our proof techniques to analyze others algorithms, e.g. SGD, SVRG [22], and SAGA [13].

**Notation.** Throughout this paper, we treat the parameters $L, \Delta, \sigma$, and $\rho$, to be specified later as global constants. Let $\|\cdot\|$ denote the Euclidean norm of a vector or spectral norm of a square matrix. Denote $p_n = \mathcal{O}(q_n)$ for a sequence of vectors $p_n$ and positive scalars $q_n$ if there is a global constant $C$ such that $|p_n| \leq Cq_n$, and $p_n = \tilde{\mathcal{O}}(q_n)$ such $C$ hides a poly-logarithmic factor of the parameters. Denote $p_n = \Omega(q_n)$ if there is a global constant $C$ such that $|p_n| \geq Cq_n$. Let $\lambda_{\min}(\mathbf{A})$ denote the least eigenvalue of a real symmetric matrix $\mathbf{A}$. For fixed $K \geq k \geq 0$, let $\mathbf{x}_{k:K}$ denote the sequence $\{\mathbf{x}^k, \dots, \mathbf{x}^K\}$. Let $[n] = \{1, \dots, n\}$ and $S$ denote the cardinality of a multi-set $\mathcal{S} \subset [n]$ of samples (a generic set that allows elements of multiple instances). For simplicity, we further denote the averaged sub-sampled stochastic estimator $\mathcal{B}_S := (1/S)\sum_{i \in \mathcal{S}} \mathcal{B}_i$ and averaged sub-sampled gradient $\nabla f_S := (1/S)\sum_{i \in \mathcal{S}} \nabla f_i$. Other notations are explained at their first appearance.

## 2 Stochastic Path-Integrated Differential Estimator: Core Idea

In this section, we present in detail the underlying idea of our Stochastic Path-Integrated Differential Estimator (SPIDER) technique behind the algorithm design. As the readers will see, such technique significantly avoids excessive access of stochastic oracle and reduces complexity, which is of independent interest and has potential applications in many stochastic estimation problems.

Let us consider an arbitrary deterministic vector quantity $Q(\mathbf{x})$. Assume that we observe a sequence $\hat{\mathbf{x}}_{0:K}$, and we want to dynamically track $Q(\hat{\mathbf{x}}^k)$ for $k = 0, 1, \dots, K$. Assume further that we have an initial estimate $\tilde{Q}(\hat{\mathbf{x}}^0) \approx Q(\hat{\mathbf{x}}^0)$, and an unbiased estimate $\boldsymbol{\xi}_k(\hat{\mathbf{x}}_{0:k})$ of $Q(\hat{\mathbf{x}}^k) - Q(\hat{\mathbf{x}}^{k-1})$ such that for each $k = 1, \dots, K$

$$\mathbb{E}\left[\boldsymbol{\xi}_k(\hat{\mathbf{x}}_{0:k}) \mid \hat{\mathbf{x}}_{0:k}\right] = Q(\hat{\mathbf{x}}^k) - Q(\hat{\mathbf{x}}^{k-1}).$$

Then we can integrate (in the discrete sense) the stochastic differential estimate as

$$\tilde{Q}(\hat{\mathbf{x}}_{0:K}) := \tilde{Q}(\hat{\mathbf{x}}^0) + \sum_{k=1}^{K} \boldsymbol{\xi}_k(\hat{\mathbf{x}}_{0:k}). \tag{2.1}$$

We call estimator $\tilde{Q}(\hat{\mathbf{x}}_{0:K})$ the *Stochastic Path-Integrated Differential EstimatoR*, or SPIDER for brevity. We conclude the following proposition which bounds the error of our estimator $\|\tilde{Q}(\hat{\mathbf{x}}_{0:K}) - Q(\hat{\mathbf{x}}^K)\|$, in terms of both expectation and high probability:

**Proposition 1.** *The martingale variance bound has*

$$\mathbb{E}\|\tilde{Q}(\hat{\mathbf{x}}_{0:K}) - Q(\hat{\mathbf{x}}^K)\|^2 = \mathbb{E}\|\tilde{Q}(\hat{\mathbf{x}}^0) - Q(\hat{\mathbf{x}}^0)\|^2 + \sum_{k=1}^{K} \mathbb{E}\|\boldsymbol{\xi}_k(\hat{\mathbf{x}}_{0:k}) - (Q(\hat{\mathbf{x}}^k) - Q(\hat{\mathbf{x}}^{k-1}))\|^2. \tag{2.2}$$

Proposition 1 can be easily concluded using the property of square-integrable martingales. Now, let $\mathcal{B}$ map any $\mathbf{x} \in \mathbb{R}^d$ to a random estimate $\mathcal{B}_i(\mathbf{x})$ such that, conditioning on the observed sequence $\mathbf{x}_{0:k}$, we have for each $k = 1, \dots, K$,

$$\mathbb{E}\left[\mathcal{B}_i(\mathbf{x}^k) - \mathcal{B}_i(\mathbf{x}^{k-1}) \mid \mathbf{x}_{0:k}\right] = \mathcal{V}^k - \mathcal{V}^{k-1}. \tag{2.3}$$

At each step $k$ let $S_*$ be a subset that samples $\mathcal{S}_*$ elements in $[n]$ with replacement, and let the stochastic estimator $\mathcal{B}_{S_*} = (1/\mathcal{S}_*)\sum_{i \in S_*} \mathcal{B}_i$ satisfy

$$\mathbb{E}\|\mathcal{B}_i(\mathbf{x}) - \mathcal{B}_i(\mathbf{y})\|^2 \leq L_{\mathcal{B}}^2 \|\mathbf{x} - \mathbf{y}\|^2, \tag{2.4}$$

and $\|\mathbf{x}^k - \mathbf{x}^{k-1}\| \leq \epsilon_1$ for all $k = 1, \dots, K$. Finally, we set our estimator $\mathcal{V}^k$ of $\mathcal{B}(\mathbf{x}^k)$ as

$$\mathcal{V}^k = \mathcal{B}_{S_*}(\mathbf{x}^k) - \mathcal{B}_{S_*}(\mathbf{x}^{k-1}) + \mathcal{V}^{k-1}.$$

Applying Proposition 1 immediately concludes the following lemma, which gives an error bound of the estimator $\mathcal{V}^k$ in terms of the second moment of $\|\mathcal{V}^k - \mathcal{B}(\mathbf{x}^k)\|$:

**Lemma 1.** *We have under the condition* (2.4) *that for all* $k = 1, \dots, K$,

$$\mathbb{E}\|\mathcal{V}^k - \mathcal{B}(\mathbf{x}^k)\|^2 \leq \frac{k L_{\mathcal{B}}^2 \epsilon_1^2}{\mathcal{S}_*} + \mathbb{E}\|\mathcal{V}^0 - \mathcal{B}(\mathbf{x}^0)\|^2. \tag{2.5}$$

It turns out that one can use SPIDER to track many quantities of interest, such as stochastic gradient, function values, zero-order estimate gradient, functionals of Hessian matrices, etc. Our proposed SPIDER-based algorithms in this paper take $\mathcal{B}_i$ as the stochastic gradient $\nabla f_i$ and the zeroth-order estimate gradient, separately.

## 3  SPIDER for Stochastic First-Order Method

In this section, we apply SPIDER to the Stochastic First-Order (SFO) method. We introduce the basic settings and assumptions in §3.1 and propose the main error-bound theorems for finding an $\epsilon$-approximate first-order stationary point in §3.2. We conclude this section with the corresponding lower-bound result in §3.3.

### 3.1  Settings and Assumptions

We first introduce the formal definition of an approximate first-order stationary point as follows.

**Definition 1.** *We call* $\mathbf{x} \in \mathbb{R}^d$ *an* $\epsilon$-*approximate first-order stationary point, or simply an FSP, if*

$$\|\nabla f(\mathbf{x})\| \leq \epsilon. \tag{3.1}$$

For our purpose of analysis, we also pose the following assumption:

**Assumption 1.** *We assume the following*

(i) *The* $\Delta := f(\mathbf{x}^0) - f^* < \infty$ *where* $f^* = \inf_{\mathbf{x} \in \mathbb{R}^d} f(\mathbf{x})$ *is the global infimum value of* $f(\mathbf{x})$;

(ii) *The component function* $f_i(\mathbf{x})$ *has an averaged* $L$-*Lipschitz gradient, i.e. for all* $\mathbf{x}, \mathbf{y}$,

$$\mathbb{E}\|\nabla f_i(\mathbf{x}) - \nabla f_i(\mathbf{y})\|^2 \leq L^2 \|\mathbf{x} - \mathbf{y}\|^2;$$

(iii) *(For online case only) the stochastic gradient has a finite variance bounded by* $\sigma^2 < \infty$, *i.e.*

$$\mathbb{E}\|\nabla f_i(\mathbf{x}) - \nabla f(\mathbf{x})\|^2 \leq \sigma^2.$$

### 3.2  Upper Bound for Finding First-Order Stationary Points

Recall that NGD has iteration update rule

$$\mathbf{x}^{k+1} = \mathbf{x}^k - \eta \frac{\nabla f(\mathbf{x}^k)}{\|\nabla f(\mathbf{x}^k)\|}, \tag{3.2}$$

where $\eta$ is a constant step size. The NGD update rule (3.2) ensures $\|\mathbf{x}^{k+1} - \mathbf{x}^k\|$ being constantly equal to the stepsize $\eta$, and might fastly escape from saddle points and converge to a second-order

**Algorithm 1** SPIDER-SFO: Input $\mathbf{x}^0$, $q$, $S_1$, $S_2$, $n_0$, $\epsilon$, and $\tilde{\epsilon}$ (For finding first-order stationary point)

1: **for** $k = 0$ to $K$ **do**
2:    **if** $\mod(k, q) = 0$ **then**
3:       Draw $S_1$ samples (or compute the full gradient for the finite-sum case), let $\mathbf{v}^k = \nabla f_{\mathcal{S}_1}(\mathbf{x}^k)$
4:    **else**
5:       Draw $S_2$ samples, and let $\mathbf{v}^k = \nabla f_{\mathcal{S}_2}(\mathbf{x}^k) - \nabla f_{\mathcal{S}_2}(\mathbf{x}^{k-1}) + \mathbf{v}^{k-1}$
6:    **end if**

7: **OPTION I**                               $\diamond$ for convergence rates in high probability
8:    **if** $\|\mathbf{v}^k\| \le 2\tilde{\epsilon}$ **then**
9:       **return** $\mathbf{x}^k$
10:   **else**
11:       $\mathbf{x}^{k+1} = \mathbf{x}^k - \eta \cdot (\mathbf{v}^k / \|\mathbf{v}^k\|)$ where $\quad \eta = \dfrac{\epsilon}{Ln_0}$
12:   **end if**

13: **OPTION II**                            $\diamond$ for convergence rates in expectation
14:   $\mathbf{x}^{k+1} = \mathbf{x}^k - \eta^k \mathbf{v}^k$ where $\quad \eta^k = \min\left(\dfrac{\epsilon}{Ln_0\|\mathbf{v}^k\|}, \dfrac{1}{2Ln_0}\right)$

15: **end for**

16: **OPTION I**: Return $\mathbf{x}^K$           $\diamond$ however, this line is *not* reached with high probability

17: **OPTION II**: Return $\tilde{\mathbf{x}}$ chosen uniformly at random from $\{\mathbf{x}^k\}_{k=0}^{K-1}$

---

stationary point [25]. We propose SPIDER-SFO in Algorithm 1, which resembles a stochastic variant of NGD with the SPIDER technique applied, so that one can maintain an estimate of $\nabla f(\mathbf{x}^k)$ at a higher accuracy under limited gradient budgets.

To analyze the convergence rate of SPIDER-SFO, let us first consider the online case for Algorithm 1. We let the input parameters be

$$S_1 = \frac{2\sigma^2}{\epsilon^2}, \qquad S_2 = \frac{2\sigma}{\epsilon n_0}, \qquad \eta = \frac{\epsilon}{Ln_0}, \qquad \eta^k = \min\left(\frac{\epsilon}{Ln_0\|\mathbf{v}^k\|}, \frac{1}{2Ln_0}\right), \qquad q = \frac{\sigma n_0}{\epsilon}, \tag{3.3}$$

where $n_0 \in [1, 2\sigma/\epsilon]$ is a free parameter to choose.[5] In this case, $\mathbf{v}^k$ in Line 5 of Algorithm 1 is a SPIDER for $\nabla f(\mathbf{x}^k)$. To see this, recall $\nabla f_i(\mathbf{x}^{k-1})$ is the stochastic gradient drawn at step $k$ and

$$\mathbb{E}\left[\nabla f_i(\mathbf{x}^k) - \nabla f_i(\mathbf{x}^{k-1}) \mid \mathbf{x}_{0:k}\right] = \nabla f(\mathbf{x}^k) - \nabla f(\mathbf{x}^{k-1}). \tag{3.4}$$

Plugging in $\mathcal{V}^k = \mathbf{v}^k$ and $\mathcal{B}_i = \nabla f_i$ in Lemma 1 of §2, we can use $\mathbf{v}^k$ in Algorithm 1 as the SPIDER and conclude the following lemma that is pivotal to our analysis.

**Lemma 2.** *Set the parameters $S_1$, $S_2$, $\eta$, and $q$ as in (3.3), and $k_0 = \lfloor k/q \rfloor \cdot q$. Then under the Assumption 1, we have*

$$\mathbb{E}\left[\|\mathbf{v}^k - \nabla f(\mathbf{x}^k)\|^2 \mid \mathbf{x}_{0:k_0}\right] \le \epsilon^2.$$

*Here we compute the conditional expectation over the randomness of $x_{(k_0+1):k}$.*

Lemma 2 shows that our SPIDER $\mathbf{v}^k$ of $\nabla f(\mathbf{x})$ maintains an error of $\mathcal{O}(\epsilon)$. Using this lemma, we are ready to present the following results for Stochastic First-Order (SFO) method for finding first-order stationary points of (1.2).

**Theorem 1** (First-order stationary point, online setting, in expectation). *Assume we are in the online case, let Assumption 1 holds, set the parameters $S_1$, $S_2$, $\eta$, and $q$ as in (3.3), and set $K = \lfloor (4L\Delta n_0)\epsilon^{-2} \rfloor + 1$. Then running Algorithm 1 with OPTION II for $K$ iterations outputs a $\tilde{\mathbf{x}}$*

*satisfying*

$$\mathbb{E}\left[\|\nabla f(\tilde{\mathbf{x}})\|\right] \le 5\epsilon. \tag{3.5}$$

*The gradient cost is bounded by $24L\Delta\sigma \cdot \epsilon^{-3} + 2\sigma^2\epsilon^{-2} + 4\sigma n_0^{-1}\epsilon^{-1}$ for any choice of $n_0 \in [1, 2\sigma/\epsilon]$. Treating $\Delta$, $L$ and $\sigma$ as positive constants, the stochastic gradient complexity is $\mathcal{O}(\epsilon^{-3})$.*

The relatively reduced minibatch size serves as the key ingredient for the superior performance of SPIDER-SFO. For illustrations, let us compare the sampling efficiency among SGD, SCSG and SPIDER-SFO in their special cases. With some involved analysis of the algorithms above, we can conclude that to ensure per-iteration sufficient decrease of $\Omega(\epsilon^2/L)$, we have

  (i)  for SGD the choice of mini-batch size is $\mathcal{O}(\sigma^2 \cdot \epsilon^{-2})$;
 (ii)  for SCSG [24] and Natasha2 [2] the mini-batch size is $\mathcal{O}(\sigma \cdot \epsilon^{-1.333})$;
(iii)  for our SPIDER-SFO, only a reduced mini-batch size of $\mathcal{O}(\sigma \cdot \epsilon^{-1})$ is needed.

Turning to the finite-sum case, analogous to the online case we let

$$S_2 = \frac{n^{1/2}}{n_0}, \qquad \eta = \frac{\epsilon}{Ln_0}, \qquad \eta^k = \min\left(\frac{\epsilon}{Ln_0\|\mathbf{v}^k\|}, \frac{1}{2Ln_0}\right), \qquad q = n_0 n^{1/2}, \tag{3.6}$$

where $n_0 \in [1, n^{1/2}]$. In this case, one computes the full gradient $\mathbf{v}^k = \nabla f_{S_1}(\mathbf{x}^k)$ in Line 3 of Algorithm 1. We conclude our second upper-bound result:

**Theorem 2** (First-order stationary point, finite-sum setting, in expectation). *Assume we are in the finite-sum case, let Assumption 1 holds, set the parameters $S_2$, $\eta^k$, and $q$ as in (3.6), set $K = \lfloor (4L\Delta n_0)\epsilon^{-2} \rfloor + 1$, and let $S_1 = [n]$, i.e. we obtain the full gradient in Line 3. Then running Algorithm 1 with OPTION II for $K$ iterations outputs a $\tilde{\mathbf{x}}$ satisfying*

$$\mathbb{E}\|\nabla f(\tilde{\mathbf{x}})\| \le 5\epsilon.$$

*The gradient cost is bounded by $n + 12(L\Delta) \cdot n^{1/2}\epsilon^{-2} + 2n_0^{-1}n^{1/2}$ for any choice of $n_0 \in [1, n^{1/2}]$. Treating $\Delta$, $L$ and $\sigma$ as positive constants, the stochastic gradient complexity is $\mathcal{O}(n + n^{1/2}\epsilon^{-2})$.*

### 3.3 Lower Bound for Finding First-Order Stationary Points

To conclude the optimality of our algorithm we need an algorithmic lower bound result [10, 37]. Consider the finite-sum case and any random algorithm $\mathcal{A}$ that maps functions $f: \mathbb{R}^d \to \mathbb{R}$ to a sequence of iterates in $\mathbb{R}^{d+1}$, with

$$[\mathbf{x}^k; i_k] = \mathcal{A}^{k-1}\left(\boldsymbol{\xi}, \nabla f_{i_0}(\mathbf{x}^0), \nabla f_{i_1}(\mathbf{x}^1), \ldots, \nabla f_{i_{k-1}}(\mathbf{x}^{k-1})\right), \quad k \ge 1, \tag{3.7}$$

where $\mathcal{A}^k$ are measure mapping into $\mathbb{R}^{d+1}$, $i_k$ is the individual function chosen by $\mathcal{A}$ at iteration $k$, and $\boldsymbol{\xi}$ is uniform random vector from $[0, 1]$. And $[\mathbf{x}^0; i_0] = \mathcal{A}^0(\boldsymbol{\xi})$, where $\mathcal{A}^0$ is a measure mapping. The lower-bound result for solving (1.2) is stated as follows:

**Theorem 3** (Lower bound for SFO for the finite-sum setting). *For any $L > 0$, $\Delta > 0$, and $2 \le n \le O\left(\Delta^2 L^2 \cdot \epsilon^{-4}\right)$, for any algorithm $\mathcal{A}$ satisfying (3.7), there exists a dimension $d = \tilde{O}\left(\Delta^2 L^2 \cdot n^2 \epsilon^{-4}\right)$, and a function $f$ satisfies Assumption 1 in the finite-sum case, such that in order to find a point $\tilde{\mathbf{x}}$ for which $\|\nabla f(\tilde{\mathbf{x}})\| \le \epsilon$, $\mathcal{A}$ must cost at least $\Omega\left(L\Delta \cdot n^{1/2}\epsilon^{-2}\right)$ stochastic gradient accesses.*

Note the condition $n \le \mathcal{O}(\epsilon^{-4})$ in Theorem 3 ensures that our lower bound $\Omega(n^{1/2}\epsilon^{-2}) = \Omega(n + n^{1/2}\epsilon^{-2})$, and hence our upper bound in Theorem 1 matches the lower bound in Theorem 3 up to a constant factor of relevant parameters, and is hence *near-optimal*. Inspired by Carmon et al. [10], our proof of Theorem 3 utilizes a specific counterexample function that requires at least $\Omega(n^{1/2}\epsilon^{-2})$ stochastic gradient accesses. Note Carmon et al. [10] analyzed such counterexample in the deterministic case $n = 1$ and we generalize such analysis to the finite-sum case $n \ge 1$.

**Remark 1.** *Note by setting $n = \mathcal{O}(\epsilon^{-4})$ the lower bound complexity in Theorem 3 can be as large as $\Omega(\epsilon^{-4})$. We emphasize that this does not violate the $\mathcal{O}(\epsilon^{-3})$ upper bound in the online case [Theorem*

*1], since the counterexample established in the lower bound depends not on the stochastic gradient variance $\sigma^2$ specified in Assumption 1(iii), but on the component number $n$. To obtain the lower bound result for the online case with the additional Assumption 1(iii), with more efforts one might be able to construct a second counterexample that requires $\Omega(\epsilon^{-3})$ stochastic gradient accesses with the knowledge of $\sigma$ instead of $n$. We leave this as a future work.*

# 4 Further Extensions

Further extensions of our SPIDER technique can be successfully applied to reduce the complexity. Limited by space, we leave the details of the following important extensions in the long version of our paper at `https://arxiv.org/abs/1807.01695` .

**Upper Bound for Finding First-Order Stationary Points, in High-Probability**   Under more stringent assumptions on the moments of stochastic gradients, our Algorithm 1 with OPTION I achieves a gradient cost of $\tilde{\mathcal{O}}(\min(n^{1/2}\epsilon^{-2}, \epsilon^{-3}))$ (note the additional polylogarithmic factor) with high probability. We detail the theorems and their proofs in the long version of our paper.

**Second-Order Stationary Point**   To find a second-order stationary point with (3.1), we can fuse our SPIDER-SFO in Algorithm 1 (OPTION I taken) with a Negative-Curvature-Search (NC-Search) iteration. In the long version of our paper (and independent of [39]), we proved rigorously that a gradient cost of $\tilde{\mathcal{O}}(\min(n^{1/2}\epsilon^{-2} + \epsilon^{-2.5}, \epsilon^{-3}))$ can be achieved under standard assumptions:

**Theorem 4** (Second-Order Stationary Point, Informal). *There exists an algorithm such that under appropriate assumptions it takes to find a $(\epsilon, \sqrt{\rho\epsilon})$-second-order stationary point, we have for the online case, when $\epsilon \leq \rho\sigma^2$ the total number of stochastic gradient computations is $\tilde{\mathcal{O}}(\epsilon^{-3})$; For the finite-sum case, when $\epsilon \leq \rho n$, the total cost of gradient access is $\tilde{\mathcal{O}}(n\epsilon^{-1.5} + n^{1/2}\epsilon^{-2} + \epsilon^{-2.5})$.*

**Zeroth-Order Stationary Point**   After the NIPS submission of this work, we propose a second application of our SPIDER technique to the stochastic zeroth-order method for problem (1.2) and achieves individual function accesses of $\mathcal{O}(\min(dn^{1/2}\epsilon^{-2}, d\epsilon^{-3}))$. To best of our knowledge, this is also the *first* time a complexity of individual function value accesses for non-convex problems has been improved to the aforementioned complexity using variance reduction techniques [22, 34].

# 5 Summary and Future Directions

We propose in this work the SPIDER method for non-convex optimization. Our SPIDER-type algorithms have update rules that are reasonably simple and achieve excellent convergence properties. However, there are still some important questions are left. For example, the lower bound results for finding a second-order stationary point are *not* complete. Specially, it is *not* yet clear if our $\tilde{\mathcal{O}}(\epsilon^{-3})$ for the online case and $\tilde{\mathcal{O}}(n^{1/2}\epsilon^{-2})$ for the finite-sum case gradient cost upper bound for finding a second-order stationary point (when $n \geq \Omega(\epsilon^{-1})$) is *optimal* or the gradient cost can be further improved, assuming both Lipschitz gradient and Lipschitz Hessian.

**Acknowledgement**   The authors would like to thank Jeffrey Z. HaoChen for his help on the numerical experiments, thank an anonymous reviewer to point out a mistake in the original proof of Theorem 1 and thank Zeyuan Allen-Zhu and Quanquan Gu for relevant discussions and pointing out references Zhou et al. [39, 40], also Jianqiao Wangni for pointing out references Nguyen et al. [28, 29], and Zebang Shen, Ruoyu Sun, Haishan Ye, Pan Zhou for very helpful discussions and comments. Zhouchen Lin is supported by National Basic Research Program of China (973 Program, grant no. 2015CB352502), National Natural Science Foundation (NSF) of China (grant nos. 61625301 and 61731018), and Microsoft Research Asia.

## Footnotes

*This work was done while Cong Fang was a Research Intern with Tencent AI Lab.

[3]Allen-Zhu [2] also obtains a gradient cost of $\tilde{\mathcal{O}}(\epsilon^{-3.25})$ to achieve a (modified and weakened) $(\epsilon, \mathcal{O}(\epsilon^{0.25}))$-approximate second-order stationary point.

[4]Here and in many places afterwards, the gradient cost also includes the number of stochastic Hessian-vector product accesses, which has similar running time with computing per-access stochastic gradient.

[5]When $n_0 = 1$, the mini-batch size is $2\sigma/\epsilon$, which is the largest mini-batch size that Algorithm 1 allows to choose.

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
