[Supplementary Material]

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

[6]If $\mathbf{x}^0 \neq \mathbf{0}$, we can simply translate the counter example as $f'(\mathbf{x}) = f(\mathbf{x} - \mathbf{x}^0)$, then $f'(\mathbf{x}^0) - \inf_{\mathbf{x} \in \mathbb{R}^d} f'(\mathbf{x}) \leq \Delta$.

[7]One of the results not included in this table is Carmon et al. [9], which finds an $\epsilon$-approximate first-order stationary point in $\mathcal{O}(n\epsilon^{-1.75})$ gradient evaluations. However, their result relies on a more stringent Hessian-Lipschitz condition, in which case a second-order stationary point can be found in similar gradient cost [21].

[8]Due to the NEON method [4, 38], nearly all existing Hessian-vector product based algorithms in stochastic optimization can be converted to ones that use stochastic gradients only.

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

# A Analysis of SPIDER

In this and next sections, we sometimes denote for brevity that $\mathbb{E}_k[\cdot] = \mathbb{E}[\cdot \mid x_{0:k}]$, the expectation operator conditional on $x_{0:k}$, for an arbitrary $k \geq 0$. We focus on the proofs of Proposition 1 and Lemma 1.

## A.1 Proof of Proposition 1

*Proof of Proposition 1.* It is straightforward to verify from the definition of $\tilde{Q}$ in (2.1) that

$$\tilde{Q}(\hat{\mathbf{x}}_{0:K}) - Q(\hat{\mathbf{x}}^K) = \tilde{Q}(\hat{\mathbf{x}}^0) - Q(\hat{\mathbf{x}}^0) + \sum_{k=1}^{K} \boldsymbol{\xi}_k(\hat{\mathbf{x}}_{0:k}) - (Q(\hat{\mathbf{x}}^k) - Q(\hat{\mathbf{x}}^{k-1}))$$

is a martingale, and hence (2.2) follows from the property of $L^2$ martingales [14]. $\square$

## A.2 Proof of Lemma 1

*Proof of Lemma 1.* For any $k > 0$, we have from Proposition 1 (by applying $\tilde{Q} = \mathcal{V}$)

$$\mathbb{E}_k \|\mathcal{V}^k - \mathcal{B}(\mathbf{x}^k)\|^2 = \mathbb{E}_k \|\mathcal{B}_{S_*}(\mathbf{x}^k) - \mathcal{B}(\mathbf{x}^k) - \mathcal{B}_{S_*}(\mathbf{x}^{k-1}) + \mathcal{B}(\mathbf{x}^{k-1})\|^2 + \|\mathcal{V}^{k-1} - \mathcal{B}(\mathbf{x}^{k-1})\|^2. \tag{A.1}$$

Then

$$\begin{aligned}
&\mathbb{E}_k \|\mathcal{B}_{S_*}(\mathbf{x}^k) - \mathcal{B}(\mathbf{x}^k) - \mathcal{B}_{S_*}(\mathbf{x}^{k-1}) + \mathcal{B}(\mathbf{x}^{k-1})\|^2 \\
&\overset{a}{=} \frac{1}{S_*} \mathbb{E} \|\mathcal{B}_i(\mathbf{x}^k) - \mathcal{B}(\mathbf{x}^k) - \mathcal{B}_i(\mathbf{x}^{k-1}) + \mathcal{B}(\mathbf{x}^{k-1})\|^2 \\
&\overset{b}{\leq} \frac{1}{S_*} \mathbb{E} \|\mathcal{B}_i(\mathbf{x}^k) - \mathcal{B}_i(\mathbf{x}^{k-1})\|^2 \\
&\overset{(2.4)}{\leq} \frac{1}{S_*} L_{\mathcal{B}}^2 \mathbb{E} \|\mathbf{x}^k - \mathbf{x}^{k-1}\|^2 \leq \frac{L_{\mathcal{B}}^2 \epsilon_1^2}{S_*},
\end{aligned} \tag{A.2}$$

where in $\overset{a}{=}$ and $\overset{b}{\leq}$, we use Eq (2.3), and $S_*$ are random sampled from $[n]$ with replacement. Combining (A.1) and (A.2), we have

$$\mathbb{E}_k \|\mathcal{V}^k - \mathcal{B}(\mathbf{x}^k)\|^2 \leq \frac{L_{\mathcal{B}}^2 \epsilon_1^2}{S_*} + \|\mathcal{V}^{k-1} - \mathcal{B}(\mathbf{x}^{k-1})\|^2. \tag{A.3}$$

Telescoping the above display for $k' = k - 1, \ldots, 0$ and using the iterated law of expectation, we have

$$\mathbb{E} \|\mathcal{V}^k - \mathcal{B}(\mathbf{x}^k)\|^2 \leq \frac{k L_{\mathcal{B}}^2 \epsilon_1^2}{S_*} + \mathbb{E} \|\mathcal{V}^0 - \mathcal{B}(\mathbf{x}^0)\|^2. \tag{A.4}$$

$\square$

## A.3 Proof of Lemma 2

*Proof of Lemma 2.* For $k = k_0$, we have

$$\begin{aligned}
&\mathbb{E}_{k_0} \|\mathbf{v}^{k_0} - \nabla f(\mathbf{x}^{k_0})\|^2 \\
&= \mathbb{E}_{k_0} \|\nabla f_{S_1}(\mathbf{x}^{k_0}) - \nabla f(\mathbf{x}^{k_0})\|^2 \leq \frac{\sigma^2}{S_1} = \frac{\epsilon^2}{2}.
\end{aligned} \tag{A.5}$$

From Line 14 of Algorithm 1 we have for all $k \geq 0$,

$$\|\mathbf{x}^{k+1} - \mathbf{x}^k\| = \min\left(\frac{\epsilon}{L n_0 \|\mathbf{v}^k\|}, \frac{1}{2 L n_0}\right) \|\mathbf{v}^k\| \leq \frac{\epsilon}{L n_0}. \tag{A.6}$$

Applying Lemma 1 with $\epsilon_1 = \epsilon/(Ln_0)$, $S_2 = 2\sigma/(\epsilon n_0)$, $K = k - k_0 \leq q = \sigma n_0/\epsilon$, we have

$$\mathbb{E}_{k_0}\|\mathbf{v}^k - \nabla f(\mathbf{x}^k)\|^2 \leq \frac{\sigma n_0 L^2}{\epsilon} \cdot \frac{\epsilon^2}{L^2 n_0^2} \cdot \frac{\epsilon n_0}{2\sigma} + \mathbb{E}_{k_0}\|\mathbf{v}^{k_0} - \nabla f(\mathbf{x}^{k_0})\|^2 \overset{(A.5)}{=} \epsilon^2, \qquad (A.7)$$

completing the proof. $\qquad\qquad\square$

## B  Proof of Expectation Results for FSP

This section devotes to the proofs of Theorems 1, 2. To prepare for them, we first conclude via standard analysis the following

**Lemma 3.** *Under the Assumption 1, setting $k_0 = \lfloor k/q \rfloor \cdot q$, we have*

$$\mathbb{E}_{k_0}\left[f(\mathbf{x}^{k+1}) - f(\mathbf{x}^k)\right] \leq -\frac{\epsilon}{4Ln_0}\mathbb{E}_{k_0}\|\mathbf{v}^k\| + \frac{3\epsilon^2}{4n_0 L}. \qquad (B.1)$$

*Proof of Lemma 3.* From Assumption 1 (ii), we have

$$\|\nabla f(\mathbf{x}) - \nabla f(\mathbf{y})\|^2 = \|\mathbb{E}_i\left(\nabla f_i(\mathbf{x}) - \nabla f_i(\mathbf{y})\right)\|^2 \leq \mathbb{E}_i\|\nabla f_i(\mathbf{x}) - \nabla f_i(\mathbf{y})\|^2 \leq L^2\|\mathbf{x} - \mathbf{y}\|^2. \text{(B.2)}$$

So $f(\mathbf{x})$ has $L$-Lipschitz continuous gradient, then

$$\begin{aligned}
f(\mathbf{x}^{k+1}) &\leq f(\mathbf{x}^k) + \langle \nabla f(\mathbf{x}^k), \mathbf{x}^{k+1} - \mathbf{x}^k \rangle + \frac{L}{2}\|\mathbf{x}^{k+1} - \mathbf{x}^k\|^2 \\
&= f(\mathbf{x}^k) - \eta^k \langle \nabla f(\mathbf{x}^k), \mathbf{v}^k \rangle + \frac{L(\eta^k)^2}{2}\|\mathbf{v}^k\|^2 \\
&= f(\mathbf{x}^k) - \eta^k\left(1 - \frac{\eta^k L}{2}\right)\|\mathbf{v}^k\|^2 - \eta^k\langle \nabla f(\mathbf{x}^k) - \mathbf{v}^k, \mathbf{v}^k \rangle \\
&\overset{a}{\leq} f(\mathbf{x}^k) - \eta^k\left(\frac{1}{2} - \frac{\eta^k L}{2}\right)\|\mathbf{v}^k\|^2 + \frac{\eta^k}{2}\|\mathbf{v}^k - \nabla f(\mathbf{x}^k)\|^2, \qquad (B.3)
\end{aligned}$$

where in $\overset{a}{\leq}$, we applied Cauchy-Schwarz inequality. Since $\eta^k = \min\left(\frac{\epsilon}{Ln_0\|\mathbf{v}^k\|}, \frac{1}{2Ln_0}\right) \leq \frac{1}{2Ln_0} \leq \frac{1}{2L}$, we have

$$\eta^k\left(\frac{1}{2} - \frac{\eta^k L}{2}\right)\|\mathbf{v}^k\|^2 \geq \frac{1}{4}\eta^k\|\mathbf{v}^k\|^2 = \frac{\epsilon^2}{8n_0 L}\min\left(2\left\|\frac{\mathbf{v}^k}{\epsilon}\right\|, \left\|\frac{\mathbf{v}^k}{\epsilon}\right\|^2\right) \overset{a}{\geq} \frac{\epsilon\|\mathbf{v}^k\| - 2\epsilon^2}{4n_0 L}, \quad (B.4)$$

where in $\overset{a}{\geq}$, we use $V(x) = \min\left(|x|, \frac{x^2}{2}\right) \geq |x| - 2$ for all $x$. Hence

$$\begin{aligned}
f(\mathbf{x}^{k+1}) &\leq f(\mathbf{x}^k) - \frac{\epsilon\|\mathbf{v}^k\|}{4Ln_0} + \frac{\epsilon^2}{2n_0 L} + \frac{\eta^k}{2}\|\mathbf{v}^k - \nabla f(\mathbf{x}^k)\|^2 \\
&\overset{\eta^k \leq \frac{1}{2Ln_0}}{\leq} f(\mathbf{x}^k) - \frac{\epsilon\|\mathbf{v}^k\|}{4Ln_0} + \frac{\epsilon^2}{2n_0 L} + \frac{1}{4Ln_0}\|\mathbf{v}^k - \nabla f(\mathbf{x}^k)\|^2. \qquad (B.5)
\end{aligned}$$

Taking expectation on the above display and using Lemma 2, we have

$$\mathbb{E}_{k_0}f(\mathbf{x}^{k+1}) - \mathbb{E}_{k_0}f(\mathbf{x}^k) \leq -\frac{\epsilon}{4Ln_0}\mathbb{E}_{k_0}\|\mathbf{v}^k\| + \frac{3\epsilon^2}{4Ln_0}. \qquad (B.6)$$

$\qquad\qquad\square$

The proof is done via the following lemma:

**Lemma 4.** *Under Assumption 1, for all $k \geq 0$, we have*

$$\mathbb{E}\|\nabla f(\mathbf{x}^k)\| \leq \mathbb{E}\|\mathbf{v}^k\| + \epsilon. \qquad (B.7)$$

*Proof.* By taking the total expectation in Lemma 2, we have

$$\mathbb{E}\|\mathbf{v}^k - \nabla f(\mathbf{x}^k)\|^2 \le \epsilon^2. \tag{B.8}$$

Then by Jensen's inequality

$$\left(\mathbb{E}\|\mathbf{v}^k - \nabla f(\mathbf{x}^k)\|\right)^2 \le \mathbb{E}\|\mathbf{v}^k - \nabla f(\mathbf{x}^k)\|^2 \le \epsilon^2.$$

So using triangle inequality

$$
\begin{aligned}
\mathbb{E}\|\nabla f(\mathbf{x}^k)\| &= \mathbb{E}\|\mathbf{v}^k - (\mathbf{v}^k - \nabla f(\mathbf{x}^k))\| \\
&\le \mathbb{E}\|\mathbf{v}^k\| + \mathbb{E}\|\mathbf{v}^k - \nabla f(\mathbf{x}^k)\| \le \mathbb{E}\|\mathbf{v}^k\| + \epsilon.
\end{aligned} \tag{B.9}
$$

This completes our proof. $\qquad\square$

Now, we are ready to prove Theorem 1.

*Proof of Theorem 1.* Taking full expectation on Lemma 3, and telescoping the results from $k = 0$ to $K - 1$, we have

$$\frac{\epsilon}{4Ln_0} \sum_{k=0}^{K-1} \mathbb{E}\|\mathbf{v}^k\| \le f(\mathbf{x}^0) - \mathbb{E}f(\mathbf{x}^K) + \frac{3K\epsilon^2}{4Ln_0} \overset{\mathbb{E}f(\mathbf{x}^K) \ge f^*}{\le} \Delta + \frac{3K\epsilon^2}{4Ln_0}. \tag{B.10}$$

Diving $\frac{\epsilon}{4Ln_0}K$ both sides of (B.10), and using $K = \lfloor \frac{4L\Delta n_0}{\epsilon^2} \rfloor + 1 \ge \frac{4L\Delta n_0}{\epsilon^2}$, we have

$$\frac{1}{K} \sum_{k=0}^{K-1} \mathbb{E}\|\mathbf{v}^k\| \le \Delta \cdot \frac{4Ln_0}{\epsilon} \frac{1}{K} + 3\epsilon \le 4\epsilon. \tag{B.11}$$

Then from the choose of $\tilde{\mathbf{x}}$, we have

$$\mathbb{E}\|\nabla f(\tilde{\mathbf{x}})\| = \frac{1}{K} \sum_{k=0}^{K-1} \mathbb{E}\|\nabla f(\mathbf{x}^k)\| \overset{(B.7)}{\le} \frac{1}{K} \sum_{k=0}^{K-1} \mathbb{E}\|\mathbf{v}^k\| + \epsilon \overset{(B.11)}{\le} 5\epsilon. \tag{B.12}$$

To compute the gradient cost, note in each $q$ iterations we access for one time $S_1$ stochastic gradients and for $q$ times of $2S_2$ stochastic gradients, and hence the cost is

$$
\begin{aligned}
\left\lceil K \cdot \frac{1}{q} \right\rceil S_1 + 2KS_2 &\overset{S_1 = qS_2}{\le} 3K \cdot S_2 + S_1 \\
&\le 3\left(\frac{4Ln_0\Delta}{\epsilon^2}\right) \frac{2\sigma}{\epsilon n_0} + \frac{2\sigma^2}{\epsilon^2} + 2S_2 \\
&= \frac{24L\sigma\Delta}{\epsilon^3} + \frac{2\sigma^2}{\epsilon^2} + \frac{4\sigma}{n_0\epsilon}.
\end{aligned} \tag{B.13}
$$

This concludes a gradient cost of $24L\Delta\sigma\epsilon^{-3} + 2\sigma^2\epsilon^{-2} + 4\sigma n_0^{-1}\epsilon^{-1}$. $\qquad\square$

*Proof of Theorem 2.* For Lemma 2, we have

$$\mathbb{E}_{k_0}\|\mathbf{v}^{k_0} - \nabla f(\mathbf{x}^{k_0})\|^2 = \mathbb{E}_{k_0}\|\nabla f(\mathbf{x}^{k_0}) - \nabla f(\mathbf{x}^{k_0})\|^2 = 0. \tag{B.14}$$

With the above display, applying Lemma 1 with $\epsilon_1 = \frac{\epsilon}{Ln_0}$, and $S_2 = \frac{n^{1/2}}{\epsilon n_0}$, $K = k - k_0 \le q = n_0 n^{1/2}$, we have

$$\mathbb{E}_{k_0}\|\mathbf{v}^k - \nabla f(\mathbf{x}^k)\|^2 \le n_0 n^{1/2} L^2 \cdot \frac{\epsilon^2}{L^2 n_0^2} \cdot \frac{\epsilon n_0}{n^{1/2}} + \mathbb{E}_{k_0}\|\mathbf{v}^{k_0} - \nabla f(\mathbf{x}^{k_0})\|^2 \overset{(A.5)}{=} \epsilon^2. \tag{B.15}$$

So Lemma 2 holds. Then from the same technique of online case, we can obtain (A.6) and (4), and (B.12). The gradient cost analysis is computed as:

$$\left\lceil K \cdot \frac{1}{q} \right\rceil S_1 + 2KS_2 \overset{S_1 = qS_2}{\leq} 3K \cdot S_2 + S_1$$

$$\leq 3\left(\frac{4Ln_0\Delta}{\epsilon^2}\right)\frac{n^{1/2}}{n_0} + n + 2S_2$$

$$= \frac{12(L\Delta) \cdot n^{1/2}}{\epsilon^2} + n + \frac{2n^{1/2}}{n_0}. \tag{B.16}$$

This concludes a gradient cost of $n + 12(L\Delta) \cdot n^{1/2}\epsilon^{-2} + 2n_0^{-1}n^{1/2}$. $\qquad\square$

## C Proof of Theorem 3 for Lower Bound

Our proof is a direct extension of Carmon et al. [10]. Before we drill into the proof of Theorem 3, we first introduce the hard instance $\hat{f}_K$ with $K \geq 1$ constructed by Carmon et al. [10].

$$\hat{f}_K(\mathbf{x}) := -\Psi(1)\Phi(x_1) + \sum_{i=2}^{K} [\Psi(-x_{i-1})\Phi(-x_i) - \Psi(x_{i-1})\Phi(x_i)], \tag{C.1}$$

where the component functions are

$$\Psi(x) := \begin{cases} 0 & x \leq \frac{1}{2} \\ \exp\left(1 - \frac{1}{(2x-1)^2}\right) & x > \frac{1}{2} \end{cases} \tag{C.2}$$

and

$$\Phi(x) := \sqrt{e} \int_{-\infty}^{x} e^{-\frac{t^2}{2}}, \tag{C.3}$$

where $x_i$ denote the value of $i$-th coordinate of $\mathbf{x}$, with $i \in [d]$. $\hat{f}_K(\mathbf{x})$ constructed by Carmon et al. [10] is a zero-chain function, that is for every $i \in [d]$, $\nabla_i f(\mathbf{x}) = 0$ whenever $x_{i-1} = x_i = x_{i+1}$. So any deterministic algorithm can only recover "one" dimension in each iteration [10]. In addition, it satisfies that : If $|x_i| \leq 1$ for any $i \leq K$,

$$\left\|\nabla \hat{f}_K(\mathbf{x})\right\| \geq 1. \tag{C.4}$$

Then to handle random algorithms, Carmon et al. [10] further consider the following extensions:

$$\tilde{f}_{K,\mathbf{B}^K}(\mathbf{x}) = \hat{f}_K\left((\mathbf{B}^K)^{\mathrm{T}}\rho(\mathbf{x})\right) + \frac{1}{10}\|\mathbf{x}\|^2 = \hat{f}_K\left(\left\langle \mathbf{b}^{(1)}, \rho(\mathbf{x})\right\rangle, \dots, \left\langle \mathbf{b}^{(K)}, \rho(\mathbf{x})\right\rangle\right) + \frac{1}{10}\|\mathbf{x}\|^2, \tag{C.5}$$

where $\rho(\mathbf{x}) = \frac{\mathbf{x}}{\sqrt{1+\|\mathbf{x}\|^2/R^2}}$ and $R = 230\sqrt{K}$, $\mathbf{B}^K$ is chosen uniformly at random from the space of orthogonal matrices $\mathcal{O}(d, K) = \{\mathbf{D} \in \mathbb{R}^{d\times K} | \mathbf{D}^{\top}\mathbf{D} = I_K \}$. The function $\tilde{f}_{K,\mathbf{B}}(\mathbf{x})$ satisfies the following:

(i)

$$\tilde{f}_{K,\mathbf{B}^K}(\mathbf{0}) - \inf_{\mathbf{x}} \tilde{f}_{K,\mathbf{B}^K}(\mathbf{x}) \leq 12K. \tag{C.6}$$

(ii) $\tilde{f}_{K,\mathbf{B}^K}(\mathbf{x})$ has constant $l$ (independent of $K$ and $d$) Lipschitz continuous gradient.

(iii) if $d \geq 52 \cdot 230^2 K^2 \log(\frac{2K^2}{p})$, for any algorithm $\mathcal{A}$ solving (1.2) with $n = 1$, and $f(\mathbf{x}) = \tilde{f}_{K,\mathbf{B}^K}(\mathbf{x})$, then with probability $1 - p$,

$$\left\| \nabla \tilde{f}_{K,\mathbf{B}^K}(\mathbf{x}^k) \right\| \geq \frac{1}{2}, \quad \text{for every } k \leq K. \tag{C.7}$$

The above properties found by Carmon et al. [10] is very technical. One can refer to Carmon et al. [10] for more details.

*Proof of Theorem 3.* Our lower bound theorem proof is as follows. The proof mirrors Theorem 2 in Carmon et al. [10] by further taking the number of individual function $n$ into account. Set

$$f_i(\mathbf{x}) := \frac{ln^{1/2}\epsilon^2}{L} \tilde{f}_{K,\mathbf{B}_i^K}(\mathbf{C}_i^{\mathrm{T}}\mathbf{x}/b) = \frac{ln^{1/2}\epsilon^2}{L}\left( \hat{f}_K\left((\mathbf{B}_i^K)^{\mathrm{T}}\rho(\mathbf{C}_i^{\mathrm{T}}\mathbf{x}/b)\right) + \frac{1}{10}\left\| \mathbf{C}_i^{\mathrm{T}}\mathbf{x}/b \right\|^2 \right), \tag{C.8}$$

and

$$f(\mathbf{x}) = \frac{1}{n}\sum_{i=1}^{n} f_i(\mathbf{x}). \tag{C.9}$$

where $\mathbf{B}^{nK} = [\mathbf{B}_1^K, \ldots, \mathbf{B}_n^K]$ is chosen uniformly at random from the space of orthogonal matrices $\mathcal{O}(d, K) = \{\mathbf{D} \in \mathbb{R}^{(d/n) \times (nK)} | \mathbf{D}^{\top}\mathbf{D} = I_{(nK)}\}$, with each $\mathbf{B}_i^K \in \{\mathbf{D} \in \mathbb{R}^{(d/n) \times (K)} | \mathbf{D}^{\top}\mathbf{D} = I_{(K)}\}$, $i \in [n]$, $\mathbf{C} = [\mathbf{C}_1, \ldots, \mathbf{C}_n]$ is an arbitrary orthogonal matrices $\mathcal{O}(d, K) = \{\mathbf{D} \in \mathbb{R}^{d \times d} | \mathbf{D}^{\top}\mathbf{D} = I_d\}$, with each $\mathbf{C}_i^K \in \{\mathbf{D} \in \mathbb{R}^{(d) \times (d/n)} | \mathbf{D}^{\top}\mathbf{D} = I_{(d/n)}\}$, $i \in [n]$. $K = \frac{\Delta L}{12ln^{1/2}\epsilon^2}$, with $n \leq \frac{144\Delta^2 L^2}{l^2\epsilon^4}$ (to ensure $K \geq 1$), $b = \frac{l\epsilon}{L}$, and $R = \sqrt{230K}$. We first verify that $f(\mathbf{x})$ satisfies Assumption 1 (i). For Assumption 1 (i), from (C.6), we have

$$f(\mathbf{0}) - \inf_{\mathbf{x} \in \mathbb{R}^d} f(\mathbf{x}) \leq \frac{1}{n}\sum_{i=1}^{n}(f_i(\mathbf{0}) - \inf_{\mathbf{x} \in \mathbb{R}^d} f_i(\mathbf{x})) \leq \frac{ln^{1/2}\epsilon^2}{L}12K = \frac{ln^{1/2}\epsilon^2}{L}\frac{12\Delta L}{12ln^{1/2}\epsilon^2} = \Delta^6.$$

For Assumption 1(ii), for any $i$, using the $\tilde{f}_{K,\mathbf{B}_i^K}$ has $l$-Lipschitz continuous gradient, we have

$$\left\| \nabla \tilde{f}_{K,\mathbf{B}_i^K}(\mathbf{C}_i^{\mathrm{T}}\mathbf{x}/b) - \nabla \tilde{f}_{K,\mathbf{B}_i^K}(\mathbf{C}_i^{\mathrm{T}}\mathbf{y}/b) \right\|^2 \leq l^2\left\| \mathbf{C}_i^{\mathrm{T}}(\mathbf{x} - \mathbf{y})/b \right\|^2, \tag{C.10}$$

Because $\|\nabla f_i(\mathbf{x}) - \nabla f_i(\mathbf{y})\|^2 = \left\| \frac{ln^{1/2}\epsilon^2}{Lb}\mathbf{C}_i\left( \nabla \tilde{f}_{K,\mathbf{B}_i^K}(\mathbf{C}_i^{\mathrm{T}}\mathbf{x}/b) - \nabla \tilde{f}_{K,\mathbf{B}_i^K}(\mathbf{C}_i^{\mathrm{T}}\mathbf{y}/b)\right) \right\|^2$, and using $\mathbf{C}_i^{\top}\mathbf{C}_i = I_{d/n}$, we have

$$\|\nabla f_i(\mathbf{x}) - \nabla f_i(\mathbf{y})\|^2 \leq \left( \frac{ln^{1/2}\epsilon^2}{L} \right)^2 \frac{l^2}{b^4}\left\| \mathbf{C}_i^{\mathrm{T}}(\mathbf{x} - \mathbf{y}) \right\|^2 = nL^2\left\| \mathbf{C}_i^{\mathrm{T}}(\mathbf{x} - \mathbf{y}) \right\|^2, \tag{C.11}$$

where we use $b = \frac{l\epsilon}{L}$. Summing $i = 1, \ldots, n$ and using each $\mathbf{C}_i$ are orthogonal matrix, we have

$$\mathbb{E}\|\nabla f_i(\mathbf{x}) - \nabla f_i(\mathbf{y})\|^2 \leq L^2\|\mathbf{x} - \mathbf{y}\|^2. \tag{C.12}$$

Then with

$$d \geq 2\max(9n^3 K^2, 12n^2 KR^2)\log\left( \frac{2n^3 K^2}{p} \right) + n^2 K \sim \mathcal{O}\left( \frac{n^2\Delta^2 L^2}{\epsilon^4}\log\left( \frac{n^2\Delta^2 L^2}{\epsilon^4 p} \right) \right),$$

from Lemma 2 of Carmon et al. [10] (or similarly Lemma 7 of Woodworth & Srebro [37] and Theorem 3 of Woodworth & Srebro [36], also refer to Lemma 5 in the end of the paper), with probability at least $1 - p$, after $T = \frac{nK}{2}$ iterations (at the end of iteration $T - 1$), for all $I_i^{T-1}$ with $i \in [d]$, if $I_i^{T-1} < K$, then for any $j_i \in \{I_i^{T-1} + 1, \ldots, K\}$, we have $\langle \mathbf{b}_{i,j_i}, \rho(\mathbf{C}_i^{\mathrm{T}}\mathbf{x}/b) \rangle \leq \frac{1}{2}$, where $I_i^{T-1}$ denotes that the algorithm $\mathcal{A}$ has called individual function $i$ with $I_i^{T-1}$ times ($\sum_{i=1}^{n} I_i^{T-1} = T$)

at the end of iteration $T - 1$, and $\mathbf{b}_{i,j}$ denotes the $j$-th column of $\mathbf{B}_i^K$. However, from (C.7), if $\langle \mathbf{b}_{i,j_i}, \rho(\mathbf{C}_i^T \mathbf{x}/b) \rangle \leq \frac{1}{2}$, we will have $\|\nabla \tilde{f}_{K,\mathbf{B}_i^K}(\mathbf{C}_i^T \mathbf{x}/b)\| \geq \frac{1}{2}$. So $f_i$ can be solved only after $K$ times calling it.

From the above analysis, for any algorithm $\mathcal{A}$, after running $T = \frac{nK}{2} = \frac{\Delta L n^{1/2}}{24l\epsilon^2}$ iterations, at least $\frac{n}{2}$ functions cannot be solved (the worst case is when $\mathcal{A}$ exactly solves $\frac{n}{2}$ functions), so

$$
\left\| \nabla f(\mathbf{x}^{nK/2}) \right\|^2 = \frac{1}{n^2} \left\| \sum_{i \text{ not solved}} \frac{ln^{1/2}\epsilon^2}{Lb} \mathbf{C}_i \nabla \tilde{f}_{K,\mathbf{B}_i^K}(\mathbf{C}_i^T \mathbf{x}^{nK/2}/b) \right\|^2
$$

$$
\overset{a}{=} \frac{1}{n^2} \sum_{i \text{ not solved}} \left\| n^{1/2}\epsilon \nabla \tilde{f}_{K,\mathbf{B}_i^K}(\mathbf{C}_i^T \mathbf{x}^{nK/2}/b) \right\|^2 \overset{(C.7)}{\geq} \frac{\epsilon^2}{8}, \qquad (C.13)
$$

where in $\overset{a}{=}$, we use $\mathbf{C}_i^\top \mathbf{C}_j = \mathbf{0}_{d/n}$, when $i \neq j$, and $\mathbf{C}_i^\top \mathbf{C}_i = I_{d/n}$. $\qquad \square$

**Lemma 5.** *Let $\{\mathbf{x}\}_{0:T}$ with $T = \frac{nK}{2}$ is informed by a certain algorithm in the form (3.7). Then when $d \geq 2\max(9n^3 K^2, 12n^3 K R^2)\log(\frac{2n^2 K^2}{p}) + n^2 K$, with probability $1 - p$, at each iteration $0 \leq t \leq T$, $\mathbf{x}^t$ can only recover one coordinate.*

*Proof.* The proof is essentially same to [10] and [36]. We give a proof here. Before the poof, we give the following definitions:

1. Let $i^t$ denotes that at iteration $t$, the algorithm choses the $i^t$-th individual function.

2. Let $I_i^t$ denotes the total times that individual function with index $i$ has been called before iteration $k$. We have $I_i^0 = 0$ with $i \in [n]$, $i \neq i^t$, and $I_{i^0}^0 = 1$. And for $t \geq 1$,

$$
I_i^t = \begin{cases} I_i^{t-1} + 1, & i = i_t. \\ I_i^{t-1}, & \text{otherwise.} \end{cases} \qquad (C.14)
$$

3. Let $\mathbf{y}_i^t = \rho(\mathbf{C}_i^T \mathbf{x}^t) = \frac{\mathbf{C}_i^T \mathbf{x}^t}{\sqrt{R^2 + \|\mathbf{C}_i^T \mathbf{x}^t\|^2}}$ with $i \in [n]$. We have $\mathbf{y}_i^t \in \mathbb{R}^{d/n}$ and $\|\mathbf{y}_i^t\| \leq R$.

4. Set $\mathcal{V}_i^t$ be the set that $\left( \bigcup_{i=1}^n \left\{ \mathbf{b}_{i,1}, \cdots \mathbf{b}_{i,\min(K,I_i^t)} \right\} \right) \bigcup \{ \mathbf{y}_i^0, \mathbf{y}_i^1, \cdots, \mathbf{y}_i^t \}$, where $\mathbf{b}_{i,j}$ denotes the $j$-th column of $\mathbf{B}_i^K$.

5. Set $\mathcal{U}_i^t$ be the set of $\left\{ \mathbf{b}_{i,\min(K,I_i^{t-1}+1)}, \cdots, \mathbf{b}_{i,K} \right\}$ with $i \in [n]$. $\mathcal{U}^t = \bigcup_{i=1}^n \mathcal{U}_i^t$. And set $\tilde{\mathcal{U}}_i^t = \left\{ \mathbf{b}_{i,\min(K,1)}, \cdots, \mathbf{b}_{i,\min(K,I_i^{t-1})} \right\}$. $\tilde{\mathcal{U}}^t = \bigcup_{i=1}^n \tilde{\mathcal{U}}_i^t$.

6. Let $\mathcal{P}_i^t \in \mathcal{R}^{(d/n) \times (d/n)}$ denote the projection operator to the span of $\mathbf{u} \in \mathcal{V}_i^t$. And let $\mathcal{P}_i^{t\perp}$ denote its orthogonal complement.

Because $\mathcal{A}^t$ performs measurable mapping, the above terms are all measurable on $\boldsymbol{\xi}$ and $\mathbf{B}^{nK}$, where $\boldsymbol{\xi}$ is the random vector in $\mathcal{A}$. It is clear that if for all $0 \leq t \leq T$ and $i \in [n]$, we have

$$
\left| \langle \mathbf{u}, \mathbf{y}_i^t \rangle \right| < \frac{1}{2}, \quad \text{for all } \mathbf{u} \in \mathcal{U}_i^t. \qquad (C.15)
$$

then at each iteration, we can only recover one index, which is our destination. To prove that (C.15) holds with probability at least $1 - p$, we consider a more hard event $\mathcal{G}^t$ as

$$
\mathcal{G}^t = \left\{ \left| \langle \mathbf{u}, \mathcal{P}_i^{(t-1)\perp} \mathbf{y}_i^t \rangle \right| \leq a \|\mathcal{P}_i^{(t-1)\perp} \mathbf{y}_i^t\| \mid \mathbf{u} \in \mathcal{U}^t \text{ (not } \mathcal{U}_i^t), i \in [n] \right\}, \quad t \geq 1, \qquad (C.16)
$$

with $a = \min \left( \frac{1}{3(T+1)}, \frac{1}{2(1+\sqrt{3T})R} \right)$. And $G^{\leq t} = \bigcap_{j=0}^t \mathcal{G}^j$.

We first show that if $\mathcal{G}^{\leq T}$ happens, then (C.15) holds for all $0 \leq t \leq T$. For $0 \leq t \leq T$, and $i \in [n]$, if $\mathcal{U}_i^t = \varnothing$, (C.15) is right; otherwise for any $\mathbf{u} \in \mathcal{U}_i^t$, we have

$$
\begin{aligned}
&\left|\langle \mathbf{u}, \mathbf{y}_i^t \rangle\right| \\
\leq \quad &\left|\left\langle \mathbf{u}, \mathcal{P}_i^{(t-1)\perp} \mathbf{y}_i^t \right\rangle\right| + \left|\left\langle \mathbf{u}, \mathcal{P}_i^{(t-1)} \mathbf{y}_i^t \right\rangle\right| \\
\leq \quad &a\|\mathcal{P}_i^{(t-1)\perp} \mathbf{y}_i^t\| + \left|\langle \mathbf{u}, \mathcal{P}_i^{t-1} \mathbf{y}_i^t \rangle\right| \leq aR + R\left\|\mathcal{P}_i^{t-1}\mathbf{u}\right\|,
\end{aligned}
\tag{C.17}
$$

where in the last inequality, we use $\|\mathcal{P}_i^{(t-1)\perp} \mathbf{y}_i^t\| \leq \|\mathbf{y}_i^{(t-1)}\| \leq R$.

If $t = 0$, we have $\mathcal{P}_i^{t-1} = \mathbf{0}_{d/n \times d/n}$, then $\left\|\mathcal{P}_i^{t-1}\mathbf{u}\right\| = 0$, so (C.15) holds. When $t \geq 1$, suppose at $t-1$, $\mathcal{G}^{\leq t}$ happens then (C.15) holds for all $0$ to $t-1$. Then we need to prove that $\|\mathcal{P}_i^{t-1}\mathbf{u}\| \leq b = \sqrt{3T}a$ with $\mathbf{u} \in \mathcal{U}_i^t$ and $i \in [n]$. Instead, we prove a stronger results: $\|\mathcal{P}_i^{t-1}\mathbf{u}\| \leq b = \sqrt{3T}a$ with all $\mathbf{u} \in \mathcal{U}^t$ and $i \in [n]$. Again, When $t = 0$, we have $\|\mathcal{P}_i^{t-1}\mathbf{u}\| = 0$, so it is right, when $t \geq 1$, by Graham-Schmidt procedure on $\mathbf{y}_i^0, \mathbf{b}_{i_0, \min(I_{i0}^0, K)}, \cdots, \mathbf{y}_i^{t-1}, \mathbf{b}_{i_{t-1}, \min(I_{i_{t-1}}^{t-1}, K)}$, we have

$$
\left\|\mathcal{P}_i^{t-1}\mathbf{u}\right\|^2 = \sum_{z=0}^{t-1} \left|\left\langle \frac{\mathcal{P}_i^{(z-1)\perp} \mathbf{y}_i^z}{\|\mathcal{P}_i^{(z-1)\perp} \mathbf{y}_i^z\|}, \mathbf{u} \right\rangle\right|^2 + \sum_{z=0,\ I_{iz}^z \leq K}^{t-1} \left|\left\langle \frac{\hat{\mathcal{P}}_i^{(z-1)\perp} \mathbf{b}_{i_z, I_{iz}^z}}{\|\hat{\mathcal{P}}_i^{(z-1)\perp} \mathbf{b}_{i_z, I_{iz}^z}\|}, \mathbf{u} \right\rangle\right|^2,
\tag{C.18}
$$

where

$$
\hat{\mathcal{P}}_i^{(z-1)} = \mathcal{P}_i^{(z-1)} + \frac{\left(\mathcal{P}_i^{(z-1)\perp} \mathbf{y}_i^z\right)\left(\mathcal{P}_i^{(z-1)\perp} \mathbf{y}_i^z\right)^{\mathrm{T}}}{\left\|\mathcal{P}_i^{(z-1)\perp} \mathbf{y}_i^z\right\|^2}.
$$

Using $\mathbf{b}_{i_z, I_{iz}^z} \perp \mathbf{u}$ for all $\mathbf{u} \in \mathcal{U}^t$, we have

$$
\begin{aligned}
&\left|\left\langle \hat{\mathcal{P}}_i^{(z-1)\perp} \mathbf{b}_{i_z, I_{iz}^z}, \mathbf{u} \right\rangle\right| \\
= \quad &\left|0 - \left\langle \hat{\mathcal{P}}_i^{(z-1)} \mathbf{b}_{i_z, I_{iz}^z}, \mathbf{u} \right\rangle\right| \\
\leq \quad &\left|\left\langle \mathcal{P}_i^{(z-1)} \mathbf{b}_{i_z, I_{iz}^z}, \mathbf{u} \right\rangle\right| + \left|\left\langle \frac{\mathcal{P}_i^{(z-1)\perp} \mathbf{y}_i^z}{\|\mathcal{P}_i^{(z-1)\perp} \mathbf{y}_i^z\|}, \mathbf{b}_{i_z, I_{iz}^z} \right\rangle \left\langle \frac{\mathcal{P}_i^{(z-1)\perp} \mathbf{y}_i^z}{\|\mathcal{P}_i^{(z-1)\perp} \mathbf{y}_i^z\|}, \mathbf{u} \right\rangle\right|.
\end{aligned}
\tag{C.19}
$$

For the first term in the right hand of (C.19), by induction, we have

$$
\left|\left\langle \mathcal{P}_i^{(z-1)} \mathbf{b}_{i_z, I_{iz}^z}, \mathbf{u} \right\rangle\right| = \left|\left\langle \mathcal{P}_i^{(z-1)} \mathbf{b}_{i_z, I_{iz}^z}, \mathcal{P}_i^{(z-1)} \mathbf{u} \right\rangle\right| \leq b^2.
\tag{C.20}
$$

For the second term in the right hand of (C.19), by assumption (C.16), we have

$$
\left|\left\langle \frac{\mathcal{P}_i^{(z-1)\perp} \mathbf{y}_i^z}{\|\mathcal{P}_i^{(z-1)\perp} \mathbf{y}_i^z\|}, \mathbf{b}_{i_z, I_{iz}^z} \right\rangle \left\langle \frac{\mathcal{P}_i^{(z-1)\perp} \mathbf{y}_i^z}{\|\mathcal{P}_i^{(z-1)\perp} \mathbf{y}_i^z\|}, \mathbf{u} \right\rangle\right| \leq a^2.
\tag{C.21}
$$

Also, we have

$$
\begin{aligned}
&\left\|\hat{\mathcal{P}}_i^{(z-1)\perp} \mathbf{b}_{i_z, I_{iz}^z}\right\|^2 \\
= \quad &\|\mathbf{b}_{i_z, I_{iz}^z}\|^2 - \left\|\hat{\mathcal{P}}_i^{(z-1)} \mathbf{b}_{i_z, I_{iz}^z}\right\|^2 \\
= \quad &\|\mathbf{b}_{i_z, I_{iz}^z}\|^2 - \left\|\mathcal{P}_i^{(z-1)} \mathbf{b}_{i_z, I_{iz}^z}\right\|^2 - \left|\left\langle \frac{\mathcal{P}_i^{(z-1)\perp} \mathbf{y}_i^z}{\|\mathcal{P}_i^{(z-1)\perp} \mathbf{y}_i^z\|}, \mathbf{b}_{i_z, I_{iz}^z} \right\rangle\right|^2 \\
\geq \quad &1 - b^2 - a^2.
\end{aligned}
\tag{C.22}
$$

Substituting (C.19) and (C.22) into (C.18), for all $\mathbf{u} \in \mathcal{U}^t$, we have

$$\left\|\mathcal{P}_i^{t-1}\mathbf{u}\right\|^2 \leq ta^2 + t\frac{(a^2+b^2)^2}{1-(a^2+b^2)}$$

$$\overset{a^2+b^2 \leq (3T+1)a^2 \leq a}{\leq} Ta^2 + T\frac{a^2}{1-a} \overset{a \leq 1/2}{\leq} 3Ta^2 = b^2. \tag{C.23}$$

Thus for (C.17), $t \geq 1$, because $\mathbf{u} \in \mathcal{U}_i^t \subseteq \mathcal{U}^t$, we have

$$\left|\langle \mathbf{u}, \mathbf{y}_i^t \rangle\right| \leq (a+b)R \overset{a \leq \frac{1}{2(1+\sqrt{3T})R}}{\leq} \frac{1}{2}. \tag{C.24}$$

This shows that if $\mathcal{G}^{\leq T}$ happens, (C.15) holds for all $0 \leq t \leq T$. Then we prove that $\mathbb{P}(\mathcal{G}^{\leq T}) \geq 1-p$. We have

$$\mathbb{P}\left((\mathcal{G}^{\leq T})^c\right) = \sum_{t=0}^{T} \mathbb{P}\left((\mathcal{G}^{\leq t})^c \mid \mathcal{G}^{<t}\right). \tag{C.25}$$

We give the following definition:

1. Denote $\hat{i}^t$ be the sequence of $i_{0:t-1}$. Let $\hat{\mathcal{S}}^t$ be the set that contains all possible ways of $\hat{i}^t$ ($|\hat{\mathcal{S}}^t| \leq n^t$).

2. Let $\tilde{\mathbf{U}}_{\hat{i}^t}^j = [\mathbf{b}_{j,1}, \cdots, \mathbf{b}_{j,\min(K,I_j^{t-1})}]$ with $j \in [n]$, and $\tilde{\mathbf{U}}_{\hat{i}^t} = [\tilde{\mathbf{U}}_{\hat{i}^t}^1, \cdots, \tilde{\mathbf{U}}_{\hat{i}^t}^n]$. $\tilde{\mathbf{U}}_{\hat{i}^t}$ is analogous to $\tilde{\mathcal{U}}^t$, but is a matrix.

3. Let $\mathbf{U}_{\hat{i}^t}^j = [\mathbf{b}_{j,\min(K,I_j^t)}; \cdots; \mathbf{b}_{j,K}]$ with $j \in [n]$, and $\mathbf{U}_{\hat{i}^t} = [\mathbf{U}_{\hat{i}^t}^1, \cdots, \mathbf{U}_{\hat{i}^t}^n]$. $\mathbf{U}_{\hat{i}^t}$ is analogous to $\mathcal{U}^t$, but is a matrix. Let $\bar{\mathbf{U}} = [\tilde{\mathbf{U}}_{\hat{i}^t}, \mathbf{U}_{\hat{i}^t}]$.

We have that

$$\mathbb{P}\left((\mathcal{G}^{\leq t})^c \mid \mathcal{G}^{<t}\right) \tag{C.26}$$

$$= \sum_{\hat{i}_0^t \in \hat{\mathcal{S}}^t} \mathbb{E}_{\boldsymbol{\xi}, \mathbf{U}_{\hat{i}_0^t}} \left(\mathbb{P}\left((\mathcal{G}^{\leq t})^c \mid \mathcal{G}^{<t}, \hat{i}^t = \hat{i}_0^t, \boldsymbol{\xi}, \mathbf{U}_{\hat{i}_0^t}\right) \mathbb{P}\left(\hat{i}^t = \hat{i}_0^t \mid \mathcal{G}^{<t}, \boldsymbol{\xi}, \mathbf{U}_{\hat{i}_0^t}\right)\right).$$

For $\sum_{\hat{i}_0^t \in \hat{\mathcal{S}}^t} \mathbb{E}_{\boldsymbol{\xi}, \mathbf{U}_{\hat{i}_0^t}} \mathbb{P}\left(\hat{i}^t = \hat{i}_0^t \mid \mathcal{G}^{<t}, \boldsymbol{\xi}, \mathbf{U}_{\hat{i}_0^t}\right) = \sum_{\hat{i}_0^t \in \hat{\mathcal{S}}^t} \mathbb{P}\left(\hat{i}^t = \hat{i}_0^t \mid \mathcal{G}^{<t}\right) = 1$, in the rest, we show that the probability $\mathbb{P}\left((\mathcal{G}^{\leq t})^c \mid \mathcal{G}^{<t}, \hat{i}^t = \hat{i}_0^t, \boldsymbol{\xi} = \boldsymbol{\xi}_0, \tilde{\mathbf{U}}_{\hat{i}_0^t} = \tilde{\mathbf{U}}_0,\right)$ for all $\boldsymbol{\xi}_0, \tilde{\mathbf{U}}_0$ is small. By union bound, we have

$$\mathbb{P}\left((\mathcal{G}^{\leq t})^c \mid \mathcal{G}^{<t}, \hat{i}^t = \hat{i}_0^t, \boldsymbol{\xi} = \boldsymbol{\xi}_0, \tilde{\mathbf{U}}_{\hat{i}_0^t} = \tilde{\mathbf{U}}_0\right) \tag{C.27}$$

$$\leq \sum_{i=1}^{n} \sum_{\mathbf{u} \in \mathcal{U}^t} \mathbb{P}\left(\left\langle \mathbf{u}, \mathcal{P}_i^{(t-1)\perp}\mathbf{y}_i^t \right\rangle \geq a\|\mathcal{P}_i^{(t-1)\perp}\mathbf{y}_i^t\| \mid \mathcal{G}^{<t}, \hat{i}^t = \hat{i}_0^t, \boldsymbol{\xi} = \boldsymbol{\xi}_0, \tilde{\mathbf{U}}_{\hat{i}_0^t} = \tilde{\mathbf{U}}_0\right).$$

Note that $\hat{i}_0^t$ is a constant. Because given $\boldsymbol{\xi}$ and $\tilde{\mathbf{U}}_{\hat{i}^t}$, under $G^{\leq t}$, both $\mathcal{P}_i^{(t-1)}$ and $\mathbf{y}_i^t$ are known. We prove

$$\mathbb{P}\left(\mathbf{U}_{\hat{i}_0^t} = \mathbf{U}_0 \mid \mathcal{G}^{<t}, \hat{i}^t = \hat{i}_0^t, \boldsymbol{\xi} = \boldsymbol{\xi}_0, \tilde{\mathbf{U}}_{\hat{i}^t} = \tilde{\mathbf{U}}_0\right) = \mathbb{P}\left(\mathbf{U}_{\hat{i}_0^t} = \mathbf{Z}_i\mathbf{U}_0 \mid \mathcal{G}^{<t}, \hat{i}^t = \hat{i}_0^t, \boldsymbol{\xi} = \boldsymbol{\xi}_0, \tilde{\mathbf{U}}_{\hat{i}^t} = \tilde{\mathbf{U}}_0\right) \text{(C.28)}$$

where $\mathbf{Z}_i \in \mathbb{R}^{d/n \times d/n}$, $\mathbf{Z}_i^{\mathrm{T}} \mathbf{Z}_i = \mathbf{I}_d$, and $\mathbf{Z}_i \mathbf{u} = \mathbf{u} = \mathbf{Z}_i^{\mathrm{T}} \mathbf{u}$ for all $\mathbf{u} \in \mathcal{V}_i^{t-1}$. In this way, $\frac{\mathcal{P}_i^{(t-1)\perp} \mathbf{u}}{\|\mathcal{P}_i^{(t-1)\perp} \mathbf{u}\|}$ has uniformed distribution on the unit space. To prove it, we have

$$
\begin{aligned}
& \mathbb{P}\left(\mathbf{U}_{\hat{i}_0^t} = \mathbf{U}_0 \mid \boldsymbol{\mathcal{G}}^{<t}, \hat{i}^t = \hat{i}_0^t, \boldsymbol{\xi} = \boldsymbol{\xi}_0, \tilde{\mathbf{U}}_{\hat{i}_0^t} = \tilde{\mathbf{U}}_0\right) \\
=\ & \frac{\mathbb{P}(\mathbf{U}_{\hat{i}_0^t} = \mathbf{U}_0, \boldsymbol{\mathcal{G}}^{<t}, \hat{i}^t = \hat{i}_0^t, \boldsymbol{\xi} = \boldsymbol{\xi}_0, \tilde{\mathbf{U}}_{\hat{i}_0^t} = \tilde{\mathbf{U}}_0)}{\mathbb{P}(\boldsymbol{\mathcal{G}}^{<t}, \hat{i}^t = \hat{i}_0^t, \boldsymbol{\xi} = \boldsymbol{\xi}_0, \tilde{\mathbf{U}}_{\hat{i}_0^t} = \tilde{\mathbf{U}}_0)} \\
=\ & \frac{\mathbb{P}(\boldsymbol{\mathcal{G}}^{<t}, \hat{i}^t = \hat{i}_0^t \mid \boldsymbol{\xi} = \boldsymbol{\xi}_0, \mathbf{U}_{\hat{i}_0^t} = \mathbf{U}_0, \tilde{\mathbf{U}}_{\hat{i}_0^t} = \tilde{\mathbf{U}}_0) p(\boldsymbol{\xi} = \boldsymbol{\xi}_0, \mathbf{U}_{\hat{i}_0^t} = \mathbf{U}_0, \tilde{\mathbf{U}}_{\hat{i}_0^t} = \tilde{\mathbf{U}}_0)}{\mathbb{P}(\boldsymbol{\mathcal{G}}^{<t}, \hat{i}^t = \hat{i}_0^t, \boldsymbol{\xi} = \boldsymbol{\xi}_0, \tilde{\mathbf{U}}_{\hat{i}_0^t} = \tilde{\mathbf{U}})}
\end{aligned}
\tag{C.29}
$$

And

$$
\begin{aligned}
& \mathbb{P}\left(\mathbf{U}_{\hat{i}_0^t} = \mathbf{Z}_i \mathbf{U}_0 \mid \boldsymbol{\mathcal{G}}^{<t}, \hat{i}^t = \hat{i}_0^t, \boldsymbol{\xi} = \boldsymbol{\xi}_0, \tilde{\mathbf{U}}_{\hat{i}_0} = \tilde{\mathbf{U}}_0\right) \\
=\ & \frac{\mathbb{P}(\boldsymbol{\mathcal{G}}^{<t}, \hat{i}^t = \hat{i}_0^t \mid \boldsymbol{\xi} = \boldsymbol{\xi}_0, \mathbf{U}_{\hat{i}_0^t} = \mathbf{U}_0, \tilde{\mathbf{U}}_{\hat{i}_0^t} = \mathbf{Z}_i \tilde{\mathbf{U}}_0) p(\boldsymbol{\xi} = \boldsymbol{\xi}_0, \mathbf{U}_{\hat{i}_0^t} = \mathbf{Z}_i \mathbf{U}_0, \tilde{\mathbf{U}}_{\hat{i}_0^t} = \tilde{\mathbf{U}}_0)}{\mathbb{P}(\boldsymbol{\mathcal{G}}^{<t}, \hat{i}^t = \hat{i}_0^t, \boldsymbol{\xi} = \boldsymbol{\xi}_0, \tilde{\mathbf{U}}_{\hat{i}_0^t} = \tilde{\mathbf{U}}_0)}
\end{aligned}
\tag{C.30}
$$

For $\boldsymbol{\xi}$ and $\bar{\mathbf{U}}$ are independent. And $p(\bar{\mathbf{U}}) = p(\mathbf{Z}_i \bar{\mathbf{U}})$, we have $p(\boldsymbol{\xi} = \boldsymbol{\xi}_0, \mathbf{U}_{\hat{i}_0^t} = \mathbf{U}_0, \tilde{\mathbf{U}}_{\hat{i}_0} = \tilde{\mathbf{U}}_0) = p(\boldsymbol{\xi} = \boldsymbol{\xi}_0, \mathbf{U}_{\hat{i}_0^t} = \mathbf{Z}_i \mathbf{U}_0, \tilde{\mathbf{U}}_{\hat{i}_0^t} = \tilde{\mathbf{U}}_0)$. Then we prove that if $\boldsymbol{\mathcal{G}}^{<t}$ and $\hat{i}^t = \hat{i}_0^t$ happens under $\mathbf{U}_{\hat{i}_0^t} = \mathbf{U}_0, \boldsymbol{\xi} = \boldsymbol{\xi}_0, \tilde{\mathbf{U}}_{\hat{i}_0^t} = \tilde{\mathbf{U}}_0$, if and only if $\boldsymbol{\mathcal{G}}^{<t}$ and $\hat{i}^t = \hat{i}_0^t$ happen under $\mathbf{U}_{\hat{i}_0^t} = \mathbf{Z}_i \mathbf{U}_0, \boldsymbol{\xi} = \boldsymbol{\xi}_0, \tilde{\mathbf{U}}_{\hat{i}_0^t} = \tilde{\mathbf{U}}_0$.

Suppose at iteration $l-1$ with $l \leq t$, we have the result. At iteration $l$, suppose $\boldsymbol{\mathcal{G}}^{<l}$ and $\hat{i}^l = \hat{i}_0^l$ happen, given $\mathbf{U}_{\hat{i}_0^t} = \mathbf{U}_0, \boldsymbol{\xi} = \boldsymbol{\xi}_0, \tilde{\mathbf{U}}_{\hat{i}_0^t} = \tilde{\mathbf{U}}_0$. Let $\mathbf{x}'$ and $(\hat{i}')^j$ are generated by $\boldsymbol{\xi} = \boldsymbol{\xi}_0, \mathbf{U}_{\hat{i}_0^t} = \mathbf{Z}_i \mathbf{U}_0, \tilde{\mathbf{U}}_{\hat{i}_0^t} = \tilde{\mathbf{U}}_0$. Because $\boldsymbol{\mathcal{G}}^{<l}$ happens, thus at each iteration, we can only recover one index until $l-1$. Then $(\mathbf{x}')^j = \mathbf{x}^j$ and $(\hat{i}')^j = \hat{i}^j$ with $j \leq l$. By induction, we only need to prove that $\boldsymbol{\mathcal{G}}^{l-1'}$ will happen. Let $\mathbf{u} \in \mathcal{U}^{l-1}$, and $i \in [n]$, we have

$$
\left| \left\langle \mathbf{Z}_i \mathbf{u}, \frac{\mathcal{P}_i^{(l-2)\perp} \mathbf{y}_i^{l-1}}{\|\mathcal{P}_i^{(l-2)\perp} \mathbf{y}_i^{l-1}\|} \right\rangle \right| = \left| \left\langle \mathbf{u}, \mathbf{Z}_i^{\mathrm{T}} \frac{\mathcal{P}_i^{(l-2)\perp} \mathbf{y}_i^{l-1}}{\|\mathcal{P}_i^{(l-2)\perp} \mathbf{y}_i^{l-1}\|} \right\rangle \right| \stackrel{a}{=} \left| \left\langle \mathbf{u}, \frac{\mathcal{P}_i^{(l-2)\perp} \mathbf{y}_i^{l-1}}{\|\mathcal{P}_i^{(l-2)\perp} \mathbf{y}_i^{l-1}\|} \right\rangle \right|, \tag{C.31}
$$

where in $\stackrel{a}{=}$, we use $\mathcal{P}_i^{(l-2)\perp} \mathbf{y}_i^{l-1}$ is in the span of $\mathcal{V}_{\hat{i}}^l \subseteq \mathcal{V}_i^{t-1}$. This shows that if $\boldsymbol{\mathcal{G}}^{<t}$ and $\hat{i}^t = \hat{i}_0^t$ happen under $\mathbf{U}_{\hat{i}_0^t} = \mathbf{U}_0, \boldsymbol{\xi} = \boldsymbol{\xi}_0, \tilde{\mathbf{U}}_{\hat{i}_0^t} = \tilde{\mathbf{U}}_0$, then $\boldsymbol{\mathcal{G}}^{<t}$ and $\hat{i}^t = \hat{i}^t$ happen under $\mathbf{U}_{\hat{i}_0^t} = \mathbf{Z}_i \mathbf{U}_0, \boldsymbol{\xi} = \boldsymbol{\xi}_0, \tilde{\mathbf{U}}_{\hat{i}_0^t} = \tilde{\mathbf{U}}_0$. In the same way, we can prove the necessity. Thus for any $\mathbf{u} \in \mathbf{U}^t$, if $\|\mathcal{P}_i^{(t-1)\perp} \mathbf{y}_i^t\| \neq 0$ (otherwise, $\left| \left\langle \mathbf{u}, \mathcal{P}_i^{(t-1)\perp} \mathbf{y}_i^t \right\rangle \right| \leq a \|\mathcal{P}_i^{(t-1)\perp} \mathbf{y}_i^t\|$ holds), we have

$$
\begin{aligned}
& \mathbb{P}\left( \left\langle \mathbf{u}, \frac{\mathcal{P}_i^{(t-1)\perp} \mathbf{y}_i^t}{\|\mathcal{P}_i^{(t-1)\perp} \mathbf{y}_i^t\|} \right\rangle \geq a \mid \boldsymbol{\mathcal{G}}^{<t}, \hat{i}^t = \hat{i}_0^t, \boldsymbol{\xi} = \boldsymbol{\xi}_0, \tilde{\mathbf{U}}_{\hat{i}_0^t} = \tilde{\mathbf{U}}_0 \right) \\
\stackrel{a}{\leq}\ & \mathbb{P}\left( \left\langle \frac{\mathcal{P}_i^{(t-1)\perp} \mathbf{u}}{\|\mathcal{P}_i^{(t-1)\perp} \mathbf{u}\|}, \frac{\mathcal{P}_i^{(t-1)\perp} \mathbf{y}_i^t}{\|\mathcal{P}_i^{(t-1)\perp} \mathbf{y}_i^t\|} \right\rangle \geq a \mid \boldsymbol{\mathcal{G}}^{<t}, \hat{i}^t = \hat{i}_0^t, \boldsymbol{\xi} = \boldsymbol{\xi}_0, \tilde{\mathbf{U}}_{\hat{i}_0^t} = \tilde{\mathbf{U}}_0 \right) \\
\stackrel{b}{\leq}\ & 2 e^{\frac{-a^2 (d/n - 2T)}{2}},
\end{aligned}
\tag{C.32}
$$

where in $\stackrel{a}{\leq}$, we use $\|\mathcal{P}_i^{(t-1)\perp} \mathbf{u}\| \leq 1$; and in $\stackrel{b}{\leq}$, we use $\frac{\mathcal{P}_i^{(t-1)\perp} \mathbf{y}_i^t}{\|\mathcal{P}_i^{(t-1)\perp} \mathbf{y}_i^t\|}$ is a known unit vector and $\frac{\mathcal{P}_i^{(t-1)\perp} \mathbf{u}}{\|\mathcal{P}_i^{(t-1)\perp} \mathbf{u}\|}$ has uniformed distribution on the unit space. Then by union bound, we have

| | Algorithm | | Online | Finite-Sum |
|---|---|---|---|---|
| First-order Stationary Point | GD / SGD | [26] | $\epsilon^{-4}$ | $n\epsilon^{-2}$ |
| | SVRG / SCSG | [3, 24, 32] | $\epsilon^{-3.333}$ | $n + n^{2/3}\epsilon^{-2}$ |
| | SPIDER-SFO | (this work) | $\epsilon^{-3}$ | $n + n^{1/2}\epsilon^{-2}$ $^\triangle$ |
| First-order Stationary Point (Hessian-Lipschitz Required) | Perturbed GD / SGD | [15, 20] | $poly(d)\epsilon^{-4}$ | $n\epsilon^{-2}$ |
| | NEON+GD / NEON+SGD | [4, 38] | $\epsilon^{-4}$ | $n\epsilon^{-2}$ |
| | AGD | [21] | N/A | $n\epsilon^{-1.75}$ |
| | NEON+SVRG / NEON+SCSG | [3, 24, 32] | $\epsilon^{-3.5}$ $(\epsilon^{-3.333})$ | $n\epsilon^{-1.5} + n^{2/3}\epsilon^{-2}$ |
| | NEON+FastCubic/CDHS | [1, 8, 35] | $\epsilon^{-3.5}$ | $n\epsilon^{-1.5} + n^{3/4}\epsilon^{-1.75}$ |
| | NEON+Natasha2 | [2, 4, 38] | $\epsilon^{-3.5}$ $(\epsilon^{-3.25})$ | $n\epsilon^{-1.5} + n^{2/3}\epsilon^{-2}$ |
| | SPIDER-SFO$^+$ | (this work) | $\epsilon^{-3}$ | $n^{1/2}\epsilon^{-2}$ $^\Theta$ |

Table 1: Comparable results on the gradient cost for nonconvex optimization algorithms that use only individual (or stochastic) gradients. Note that the gradient cost hides a poly-logarithmic factors of $d$, $n$, $\epsilon$. For clarity and brevity purposes, we record for most algorithms the gradient cost for finding an $(\epsilon, \mathcal{O}(\epsilon^{0.5}))$-approximate second-order stationary point. For some algorithms we added in a bracket underneath the best gradient cost for finding an $(\epsilon, \mathcal{O}(\epsilon^{\alpha}))$-approximate second-order stationary point among $\alpha \in (0, 1]$, for the fairness of comparison.
$\triangle$: we provide lower bound for this gradient cost entry.
$\Theta$: this entry is for $n \geq \Omega(\epsilon^{-1})$ only, in which case SPIDER-SFO$^+$ outperforms NEON+FastCubic/CDHS.

$\mathbb{P}\left(\left(\mathcal{G}^{\leq t}\right)^c \mid \mathcal{G}^{<t}\right) \leq 2(n^2 K)e^{\frac{-a^2(d/n - 2T)}{2}}$. Thus

$$
\begin{aligned}
\mathbb{P}\left(\left(\mathcal{G}^{\leq T}\right)^c\right) \quad &\leq \quad 2(T+1)n^2 K \exp\left(\frac{-a^2(d/n - 2T)}{2}\right) \\
&\overset{T=\frac{nK}{2}}{\leq} \quad 2(nK)(n^2 K) \exp\left(\frac{-a^2(d/n - 2T)}{2}\right). \quad \text{(C.33)}
\end{aligned}
$$

Then by setting

$$
\begin{aligned}
d/n \quad &\geq \quad 2\max(9n^2 K^2, 12nKR^2)\log(\frac{2n^3 K^2}{p}) + nK \\
&\geq \quad 2\max(9(T+1)^2, 2(2\sqrt{3T})^2 R^2)\log(\frac{2n^3 K^2}{p}) + 2T \\
&\geq \quad 2\max(9(T+1)^2, 2(1+\sqrt{3T})^2 R^2)\log(\frac{2n^3 K^2}{p}) + 2T \\
&\geq \quad \frac{2}{a^2}\log(\frac{2n^3 K^2}{p}) + 2T, \quad \text{(C.34)}
\end{aligned}
$$

we have $\mathbb{P}\left(\left(\mathcal{G}^{\leq T}\right)^c\right) \leq p$. This ends proof.

$\square$

## D  Comparison with Concurrent Works

We detail our main result for applying SPIDER to first-order methods in the list below:

(i) For the problem of finding an $\epsilon$-approximate first-order stationary point, under Assumption 1 our results indicate a gradient cost of $\mathcal{O}(\min(\epsilon^{-3}, n^{1/2}\epsilon^{-2}))$ which supersedes the best-known convergence rate results for stochastic optimization problem (1.2) [Theorems 1 and 2]. Before this work, the best-known result is $\mathcal{O}\left(\min(\epsilon^{-3.333}, n^{2/3}\epsilon^{-2})\right)$, achieved by Allen-Zhu & Hazan

Figure 1: Left panel: gradient cost comparison for finding an $\epsilon$-approximate first-order stationary point. Right panel: gradient cost comparison for finding an $(\epsilon, \mathcal{O}(\epsilon^{0.5}))$-approximate second-order stationary points (note we assume Hessian Lipschitz condition). Both axes are on the logarithmic scale of $\epsilon^{-1}$.

[3], Reddi et al. [32] in the finite-sum case and Lei et al. [24] in the online case, separately. Moreover, such a gradient cost achieves the algorithmic lower bound for the finite-sum setting [Theorem 3].

(ii) For the problem of finding $(\epsilon, \delta)$-approximate second-order stationary point $x$, the gradient cost is $\tilde{\mathcal{O}}(\epsilon^{-3} + \epsilon^{-2}\delta^{-2} + \delta^{-5})$ in the online case and $\tilde{\mathcal{O}}(n^{1/2}\epsilon^{-2} + n^{1/2}\epsilon^{-1}\delta^{-2} + \epsilon^{-1}\delta^{-3} + \delta^{-5} + n)$. In the classical definition of second-order stationary point where $\delta = \mathcal{O}(\epsilon^{0.5})$, such gradient cost is simply $\mathcal{O}(\epsilon^{-3})$ in the online case. In comparison, to the best of our knowledge the best-known results only achieve a gradient cost of $\mathcal{O}(\epsilon^{-3.5})$ under similar assumptions [2, 4, 31, 35, 39].

We summarize the comparison with concurrent works that solve (1.2) under similar assumptions in Table 1. In addition, we provide Figure 1 which draws the gradient cost against the magnitude of $n$ for both an approximate stationary point.[7] For simplicity, we leave out the complexities of the algorithms that has Hessian-vector product access and only record algorithms that use stochastic gradients only.[8] Specifically, the yellow-boxed complexities in Table 1 achieved by NEON+FastCubic/CDHS [4] and nonconvex AGD [21] for finding approximate second-order stationary points in the finite-sum case using momentum technique, are the only result that has *not* been outperformed by our SPIDER-SFO$^+$ algorithm in certain parameter regimes ($n \leq \mathcal{O}(\epsilon^{-1})$ in this case).

# E Experiments

We use SPIDER-SFO to optimize a synthetic function and a neural network. The synthetic function is similar to [35], with the stochasticity coming from a random coordinate shift instead of the noise on the gradient oracle. In the experiment of neural networks, we train a fully connected network on MNIST dataset.

## E.1 Synthetic function

We optimize the following function:

$$\mathbb{E}_{a,b}[w(x_1 - a) + 10(x_2 - b)^2],$$

where $a$ and $b$ are independently drawn from $\mathcal{N}(0, 0.1)$. $w(\cdot)$ is a W-shaped scalar function with a local maximum at the origin and two local minima on either side. We defer the exact form of $w(\cdot)$

Figure 2: Results on synthetic non-convex optimization problem.

to appendix. We initialize with $x_1 = x_2 = 0$, and use SPIDER-SFO to optimize the function in an online manner. We compare SPIDER with SGD and SVRG. For each algorithm, we tune the learning rate with a grid search and plot the optimal choice. The hyper-parameters and grid search settings can be found later.

We plot the relationship between function value and number of gradient oracles called in Figure 2. It can be seen that SPIDER escapes saddle point and converges to the global minimum faster than both SGD and SVRG.

**Exact form of $w$:** The function $w(\cdot)$ in the experiments of synthetic experiment is defined as [35]

$$
w(x) = \begin{cases}
\sqrt{\epsilon}(x + (L+1)\sqrt{\epsilon})^2 - \frac{1}{3}(x + (L+1)\sqrt{\epsilon})^3 - \frac{1}{3}(3L+1)\epsilon^{3/2}, & x \le -L\sqrt{\epsilon}; \\[2mm]
\epsilon x + \frac{\epsilon^{3/2}}{3} & -L\sqrt{\epsilon} < x \le -\sqrt{\epsilon}; \\[2mm]
-\sqrt{\epsilon}x^2 - \frac{x^3}{3}, & -\sqrt{\epsilon} < x \le 0; \\[2mm]
-\sqrt{\epsilon}x^2 + \frac{x^3}{3}, & 0 < x \le \sqrt{\epsilon}; \\[2mm]
-\epsilon x + \frac{\epsilon^{3/2}}{3} & \sqrt{\epsilon} < x \le L\sqrt{\epsilon}; \\[2mm]
\sqrt{\epsilon}(x - (L+1)\sqrt{\epsilon})^2 - \frac{1}{3}(x - (L+1)\sqrt{\epsilon})^3 - \frac{1}{3}(3L+1)\epsilon^{3/2}, & L\sqrt{\epsilon} \le x.
\end{cases}
\tag{E.1}
$$

We set $\epsilon = 0.01$ and $L = 5$ in the experiments.

**Hyper-parameters:** For SGD, we use minibatch of size 100. For SPIDER-SFO, we set $s1 = 1000$, $s2 = 100$ and $q = 10$. We set the $\tilde{\epsilon}$ in the algorithm as $10^{-10}$. For SVRG, we set the larger batch size with 1000 and the smaller batch size as 100, and update the large batch gradient every 10 iterations.

**Learning rate:** For each algorithm, we grid search the learning rate from set

$$\{0.1, 0.3, 0.01, 0.03, 0.001, 0.003, 0.0001, 0.0003\}$$

and choose the best one.

Figure 3: Results on MNIST classification.

## E.2 Neural network

In addition to the synthetic function, we also train a two hidden-layer fully connected neural network for classification on the MNIST dataset. Results are presented in Figure 3. In addition to SPIDER and SGD, we also compare with SCSG [24], an online version of SVRG. For both SPIDER and SGD, we set the larger batch size as $512$ and the smaller batch size as $32$.

In the real experiment on MNIST, we find that SPIDER only achieves a slight improvement. We leave designing a practical version of SPIDER in the near future.

All the experiments with neural networks are implemented with PyTorch. Implementation details are as follows:

**Dataset:** We use MNIST dataset with $50000$ training data. Each data point is a picture with $28 \times 28$ pixels with one of $10$ labels. We train a network to minimize the cross-entropy loss.

**Network architecture:** We use a 2 hidden-layer fully connected neural network, where each hidden layer contains $512$ neurons. The weights of the network are initialized randomly with normal distribution $\mathcal{N}(0, 0.01)$.

**Hyper-parameters:** For SGD, we use minibatch of size $512$. For SPIDER-SFO, we set $s1 = 512$, $s2 = 32$ and $q = 16$. For SCSG, we set the larger batch size with $512$ and the smaller batch size as $32$.

**Learning rate:** For each algorithm, we grid search the learning rate from set

$$\{0.1, 0.3, 0.01, 0.03\}$$

and choose the best one.