[Reviews · NeurIPS 2018]

Reviewer 1



I enjoyed reading this paper. In particular, I liked the compact comparison in Table 1. I believe that the results are interesting. Unfortunately, the authors have not tested their algorithm experimentally with some synthetic cases, for instance. === AFTER REBUTTAL === I thank the authors for including some experiments in the rebuttal. I raise my evaluation from 6 to 7, mainly because I reconsidered the importance and impact of the theoretical results.

Reviewer 2



Summary: Non-convex optimization has gotten a lot of attention these days, and this paper proposes a new technique in that field, called SPIDER (Stochastic Path-Integrated Differential EstimatoR) mainly to reduce the gradient computation. The paper particularly employ SPIDER in two scenarios: 1) SPIDER-SFO for first-order stationary point and 2) SPIDER-SSO for the second-order. It resulted in a very light gradient computation cost (record-breaking according to authors). Authors provide various theoretical analyses on the algorithms. Quality: The technical aspects of the paper looks solid with massive introduction and related work, followed by theoretical analyses. I particularly enjoyed reading Section 2. where the main results were sketched in a very high-level. It certainly provided a good overview for this paper. One thing concerns me is lack of experiments in the paper, which is already mentioned in the Future Directions section; however, I do think the lack of empirical analysis weakens paper a lot. I hope to see at least some of them in the paper. Clarity: Overall, the paper was quite clear to follow. Author may revise a bit of the introduction section because it was quite overwhelming to see this gigantic section on the first read without any subsections and more organized formatting. Sometimes it’s better to use bullet points and tables to make a side-by-side comparisons and emphasize the key points. Minor point: How about we restructure 4.2 as a dedicated section similar to SFO? or change the Section 4 to state it includes both SFO and SSO? Originality: The paper contains a solid novel contributions. As authors claimed in related work section, especially in the middle part of the page 3, SPIDER differs from other approaches and it’s quite extensively discussed, which resulted in a solid speed up for computing gradients. Significance: Although paper’s contribution is potentially significant, it was not empirically proven, and thus weakens the value of the paper. It is because often times theoretical bounds does not necessarily guarantee improvements in the real examples. Conclusion: I have a mixed feeling on this paper. Even though the paper showed solid contents on its theoretical supports, I feel the lack of empirical support makes me hesitate to recommend this paper. I’m happy to revise my review based on what authors/ and other reviewers comment. ------------------------------------------------------------------------------------------------- Post rebuttal comment: I thank authors for the experimental supports and filling the gap of the proof. I think the paper is in a good shape. Thank you!

Reviewer 3



Update: I have read the reviewer's response and I believe their proposed corrections makes the main result correct. Therefore, I now strongly support accepting this paper. I would be quite interested in seeing high-probability analysis that shows how (with subgaussian assumptions) to avoid the need to randomly pick a past iterate, which seems to be something no sane method would do. --- The paper proposes a stochastic optimization scheme for smooth nonconvex functions. The scheme combines fixed-norm stochastic gradient steps with a novel approach to unbiased gradient estimation, based on summing estimates of the increments in the gradient and periodically resetting the summation using an accurate gradient estimate. The paper provides an analysis of this scheme, claiming rates of convergence to stationarity that improve upon the best known rates for both the fully stochastic and the finite-sum settings. Additionally, the algorithm is extended to guarantee convergence to approximate second-order stationary points, and a matching lower bound is given in the finite-sum setting. In my opinion the proposed algorithm is original and elegant, and --- if the claimed rates of convergence were proven --- would make an excellent contribution to the active line of work on efficient scalable algorithms for nonconvex optimization. Unfortunately, I believe that the proof of the main result in the paper has fairly significant gaps, and cannot be accepted in its current state. The above-mentioned gaps stem from incorrect statements on conditional expectations. In Lemma 6, everything is conditional on the event \|v^k\| \ge 2\epsilon. Hence, the transition from Eq. (25) to Eq. (26) is based on the claim \E [ \|v^k - \nabla f(x^k) \|^2 | \|v^k\| \ge 2\epsilon ] \le \epsilon^2. However, Lemma 2 does not, and in all likelihood cannot, establish such bound on the above conditional expectation. This issue repeats in Lemma 7, where in Eq. (28) the authors (implicitly) assert that \E [ \|v^k - \nabla f(x^k) \|^2 | \|v^k\| \le 2\epsilon ] \le \epsilon^2, which is just as problematic. Worse, when Lemma 6 is applied in the proof of Theorem 1, the expectation bound is actually stated conditional on the event {\|v^i\| \ge 2\epsilon for every i \in [K]}, so a valid proof of Lemma 6 would need to show \E [ \|v^k - \nabla f(x^k) \|^2 | \|v^i\| \ge 2\epsilon, i=1,...,K ] \le \epsilon^2, which is very unlikely to be true or provable. A similar problem exists with the application of Lemma 7. I encourage the authors to try and fix their analysis: in my opinion doing so will be challenging but by no means a lost cause. However, some structural changes in the algorithm are likely necessary. Namely, the algorithm in its current form promises small gradient (in expectation) at its last iterate, while all other comparable methods only give a guarantee on a randomly sampled iterate --- it is likely that a correct version of the proposed method will have to do that as well. One possible approach to sidestepping difficulties with conditioning would be to add a sub-Gaussianity assumption on the noise and appeal to martingale concentration results. Below are two additional comments: - What about the important special case where f is convex? Does the method attain the optimal eps^{-2} convergence rate to eps-approximate global minimia? What happens when f is non-smooth? - The parameter n_0 doesn't represent a batch size, and so its name is somewhat confusing. I think it would be better to leave \mc{S}_2 as a free parameter (ranging from 1 to 2\sigma/\epsilon or n^{1/2} in the stochastic and finite-sum cases respectively). Then we have q = \mc{S}_1 / \mc{S}_2 which I think is much clearer and more intuitive.